# PROBING TO REFINE: REINFORCEMENT DISTILLATION OF LLMS VIA EXPLANATORY INVERSION

**Zhen Tan**[1][*] **Chengshuai Zhao**[1] **Song Wang**[2] **Jundong Li**[2] **Tianlong Chen**[3] **Huan Liu**[1]

[1]Arizona State University    [2]University of Virginia    [3]University of North Carolina at Chapel Hill

{ztan36,czhao93,huanliu}@asu.edu {sw3wv,jundong}@virginia.edu tianlong@cs.unc.edu

## ABSTRACT

Distilling robust reasoning capabilities from large language models (LLMs) into smaller, computationally efficient student models remains an unresolved challenge. Despite recent advances, distilled models frequently suffer from superficial pattern memorization and subpar generalization. To overcome these limitations, we introduce a novel distillation framework that moves beyond simple mimicry to instill a deeper conceptual understanding. Our framework features two key innovations. *First*, to address pattern memorization, Explanatory Inversion (EI) generates targeted "explanatory probes" that compel the student to articulate the underlying logic behind an answer, rather than just memorizing it. *Second*, to improve generalization, Explanatory GRPO (`EXGRPO`) uses a reinforcement learning algorithm with a novel Dialogue Structure Utility Bonus, which explicitly rewards the student for maintaining a coherent reasoning process across these probes. Extensive evaluations on 12 datasets demonstrate significant improvements. Using Gemma-7b as the student model, our method yields an average **20.39%** increase over zero-shot performance and a **6.02%** improvement over the state-of-the-art distillation baselines. Moreover, models distilled with our method show remarkable training efficiency (e.g., surpassing vanilla fine-tuning with **10-25%** training data) and strong generalization to out-of-distribution tasks. Implementation is released at https://github.com/Zhen-Tan-dmml/ExGRPO.git.

## 1 INTRODUCTION

Large Language Models (LLMs) have demonstrated remarkable capabilities in complex reasoning tasks, frequently leveraging techniques such as Chain-of-Thought (CoT) prompting to articulate step-by-step reasoning processes (Wei et al., 2022; Kojima et al., 2022). However, distilling these sophisticated reasoning abilities into smaller, more computationally efficient student models remains a significant open challenge (Hinton et al., 2015; Hsieh et al., 2023; Li et al., 2025d). The difficulties include exposure bias due to fixed supervision targets (Xu et al., 2019), loss of multimodal (Tian et al., 2025) or long-context reasoning capabilities (Yan et al., 2025; Yeo et al., 2025), sensitivity to distribution shifts (Yang et al., 2024c), and inefficiencies in capturing nuanced preference or alignment signals from teachers (Gu et al., 2025).

More recently, studies show that LLMs suffer from generalization limitations, exemplified by challenges like compositional generalization (Yang et al., 2024b) and out-of-distribution generalization (Zhao et al., 2025). Our work goes a step further by pioneering the finding that generalization limitations are not only present but are **amplified in distilled LLMs**, as illustrated in Figure 1(a). Existing knowledge distillation (KD) methods (Xu et al., 2024b; Yang et al., 2024a) rely heavily on Supervised Fine-Tuning (SFT), forcing the student to mimic fixed teacher outputs. Such training encourages superficial pattern memorization that breaks down under even mild distribution shifts (Chu et al., 2025). The reversal curse (Guo et al., 2024; Berglund et al., 2023) exemplifies this issue (Figure 1(b)): while distilled models may correctly solve forward problems (e.g., $5 - 2 = 3$), they often fail on the inverse (e.g., $3 + 2 = 5$).

Although not explicitly studying this challenge, recent KD methods (SU et al., 2025; Zhang et al., 2025c) have attempted to mitigate such generalization issues by employing "reverse thinking" (Deng & Li, 2022) to augment the training data, particularly on creating symmetric samples from a known

---

[*]Corresponding Author

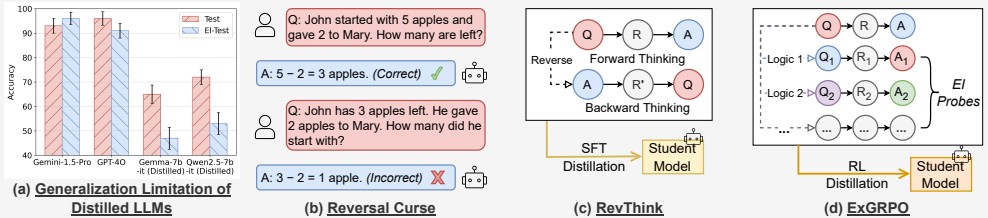

Figure 1: (a) Distilled LLMs often exhibit generalization limitations compared to teacher models (e.g., Gemini-1.5-Pro *v.s.* smaller distilled models on Test *v.s.* EI-Test set, which is the augmented version of the test set using Gemini-1.5-Pro and Explanatory Inversion (EI)). See more experimental details in Appendix D. (b) This is exemplified by the reversal curse, where a model correctly solves a forward problem (e.g., 5-2=3) but fails its inverse. (c) Prior "Reverse Thinking" approaches, like RevThink (Chen et al., 2025a), attempt A-to-Q reasoning. (d) Our ExGRPO method enhances distillation by using EI probes to challenge and refine student models via RL.

solution back to the problem. As illustrated in Figure 1(c), frameworks like RevThink (Chen et al., 2025a) train models on explicitly generated backward questions and reasoning, essentially learning the A-to-Q path (e.g., training on "End with 3, Add 2 → Started with 5" for the example above). While beneficial, these methods still encourage the student to internalize another directional mapping, rather than developing a conceptual grasp of the underlying mathematical relationships (e.g., addition-subtraction duality). As a result, the model may learn to invert outputs mechanically without cultivating the deeper understanding needed for robust generalization.

We tackle the generalization limitations from two perspectives. From the data perspective, we move beyond A-to-Q reversals and draw inspiration from the concept of **Explanatory Inversion** (EI), rooted in cognitive science's emphasis on explanation-seeking for true understanding (Searle, 1998; Potochnik & Sanches de Oliveira, 2020; Woodward, 2005). Genuine comprehension transcends rote memorization, involving recognition of understanding its context, the underlying principles, and its implications (Ausubel, 1966). Considering again the apple problem, a deeper understanding involves grasping why subtraction is used, the inverse relationship with addition, and the conditions under which the calculation holds. EI operationalizes this depth of inquiry. As dipicted in Figure 1 (d), our EI technique generates targeted "explanatory probes" ($Q_i^{aug}$) related to the original reasoning ($Q \to A, R_T$). For the apple example ($5 - 2 = 3$), probes might include: "Why is subtraction the correct operation here?", "What operation would find the starting number if you know the end number (3) and the amount given away (2)?", "Does the order '$5 - 2$' matter? What about '$2 - 5$'?", or "What if Mary gave apples to John instead?". By learning from the teacher model to answer the diverse set of probes, which require understanding the logic rather than just reversing sequences, the student model is compelled to build a richer conceptual model of the task, thereby promoting generalizability.

Furthermore, from the *optimization perspective*, we propose to leverage Reinforcement Learning (RL) to tackle the intrinsic limitation of SFT. RL promotes internalization by encouraging exploration and developing more generalizable strategies (Chu et al., 2025; Guo et al., 2025a). Although RL is often used for post-training of large models (Guo et al., 2025a), we propose the **Explanatory GRPO** (ExGRPO) algorithm, as a core component of the *distillation process itself*. The goal is not only to learn from a reward *detached* from the teacher, but to use RL to guide the student towards internalizing the complex reasoning structures *revealed* by the teacher through EI. ExGRPO leverages the outcome reward and crucially incorporates the novel **Dialogue Structure Utility Bonus**. This bonus is designed to reward the student for demonstrating a coherent understanding across the sequence of explanatory probes derived from the teacher's logic, thereby optimizing for a deeper form of knowledge transfer than simple imitation through EI.

Our main contributions include: (***i***) We pioneer the identification of the amplified generalization limitation of distilled LLMs. (***ii***) We introduce EI, a cognitively inspired reasoning augmentation technique that systematically probes and reinforces the student model's understanding of the underlying logic, moving beyond superficial pattern memorization. (***iii***) We propose ExGRPO, a novel reinforcement learning-based distillation algorithm that leverages a specifically designed *Dialogue Structure Utility Bonus* to promote internalization and coherent reasoning processes explicitly revealed by EI. (***iv***) Comprehensive experiments across 12 datasets demonstrate significant improvements in reasoning capability (avg. 20.39% over zero-shot, 6.02% over state-of-the-art KD baselines), robustness, sample & token efficiency, and out-of-distribution (OOD) generalization for distilled models.

## 2    RELATED WORK

We briefly discuss the most critical related works here. A detailed version is given in Appendix A.

**Knowledge Distillation for LLMs.** Knowledge Distillation (KD) is a popular technique for creating smaller, efficient student LLMs from larger teacher models (Xu et al., 2024b). While KD is widely used for various model refinement tasks, such as self-distillation (Yang et al., 2024c), we focus on the canonical setting where a small, efficient student model distills knowledge from a strong, large teacher. While early methods focused on aligning logits (Sanh et al., 2019), modern approaches distill reasoning capabilities by training students on teacher-generated outputs like Chain-of-Thought (CoT) rationales (Hsieh et al., 2023; Mukherjee et al., 2023). Our work extends this line of research by using novel Explanatory Inversion (EI) probes to elicit and distill more robust reasoning.

**Challenges of Distilled LLMs.** Despite progress, distilled LLMs often suffer from limited generalization due to challenges like exposure bias, teacher-student discrepancies, and an inability to internalize complex reasoning (Xu et al., 2019; Chen et al., 2025d). This can lead to shallow reasoning and logical inconsistencies. While methods like RevThink (Chen et al., 2025a) address specific failure modes like the reversal curse by augmenting data, our work aims to foster a deeper, more generalizable understanding beyond rote pattern matching.

**Rule-based RL for LLMs.** Reinforcement Learning (RL) is increasingly used to enhance LLM generalizability (Schulman et al., 2017; Rafailov et al., 2023). While rule-based methods like GRPO can elicit reasoning from outcome-only rewards (Guo et al., 2025a), they are ill-suited for distillation. GRPO optimizes the policy using only final answer correctness as the reward signal. It does not incorporate any intermediate reasoning steps or teacher-generated traces, making it unable to supervise the student's reasoning behavior. Our Explanatory GRPO (ExGRPO) algorithm overcomes this by introducing a novel Dialogue Structure Utility (DSU) reward, which enables the teacher model to explicitly supervise coherent reasoning across multi-turn dialogues while encouraging exploration.

## 3    REINFORCEMENT DISTILLATION VIA EXPLANATORY INVERSION

This work aims to improve the limited generalizability of distilled LLMs. We achieve this by first using **Explanatory Inversion** (EI) to systematically probe the student model's understanding with targeted challenges derived from teacher reasoning. Subsequently, our novel **Explanatory GRPO** (ExGRPO) algorithm refines the student model, using these interactions to promote not only final answer accuracy but also the coherent processing of explanatory dialogues, effectively enhancing the distillation of nuanced reasoning. All the hyperparameters are detailed in Appendix I.

### 3.1    EXPLANATORY INVERSION: CRAFTING PROBES FOR DEEPER REASONING

The core of our data augmentation strategy is **Explanatory Inversion (EI)**, a technique designed to generate a diverse set of "explanatory probes" that compel student models to move beyond superficial pattern matching towards deeper conceptual understanding. Given an original distillation data point containing question-answer pair $D(Q) = (Q, A, R_T)$ where $R_T = (s_1, s_2, \ldots, s_j, \ldots, s_T)$ its corresponding teacher-generated CoT reasoning and $s_j$ is a reasoning step, EI aims to deconstruct, challenge, and explore the underlying logic, assumptions, and principles within $R_T$. EI systematically applies $N(= 10$ in this paper) distinct categories of transformation rules, $F = \{f_1, \ldots, f_N\}$, to $(Q, A, R_T)$ to generate a spectrum of new probing questions $Q_i^{\text{aug}} = f_i(Q, A, R_T)$. The specific formulation of each $f_i$ involves template-based transformations. This process yields a set of up to $N$ augmented data tuples for each original example: $D_{cand}(Q) = F(D(Q)) = \{d_k\}_{i=1}^N = \{(Q_i^{\text{aug}}, A_i^{\text{aug}}, R_{T,i}^{\text{aug}})\}_{i=1}^N$. These tuples serve as candidate augmentations. The choices of the EI probes are based on diverse cognitive findings (Keil, 2006; Sloman & Sloman, 2009; Machamer et al., 2000; Byrne, 2007; Gentner, 1983; Klahr & Dunbar, 1988; Lave & Wenger, 1991; Newell et al., 1972; Zacks et al., 2007; Chi et al., 1989). Each probe is engineered to contain a rule $f_i$ to assess and cultivate specific facets of reasoning. For brevity, we present two adopted rules below, which transform an original task by creating a counterfactual scenario (R1) or demanding a more detailed justification of its premise (R2). The full lists with examples are detailed in Appendix E and N.

> **Explanatory Inversion Rule Examples**
>
> **R1.  Counterfactual Scenario Generation** ($f_1$)**:** Creates probes exploring outcomes if a premise were altered. This tests understanding of conditional dependencies and logical consequences (Byrne, 2007).
> - **Original Task: Premise**: "All passengers and crew survived the crash." **Hypothesis**: "No children died." **Label**: Entailment.
> - **Augmented Probe:** How would the entailment change if the premise stated that "**most**" rather than "**all**" passengers survived?

> **R2. Explanatory Challenge** ($f_3$): Poses questions that demand justification for a specific logical transition. This encourages explicit articulation of granular inferences and fosters self-reflection (Chi et al., 1989).
> - **Original Task: Premise**: "All passengers and crew survived." Is it reasonable to conclude that no children died?
> - **Augmented Probe: Explain** the logical steps connecting the premise "**all passengers and crew survived**" to the conclusion that "**no children were killed**."

### 3.2 STAGE 1: DATA CURATION

The initial stage of our framework involves curating a high-quality dataset from the EI-generated probes and then performing Supervised Fine-Tuning (SFT) to warm up the student model to the diverse reasoning challenges generated by EI, providing an initial alignment with teacher insights. We carefully study the effect of SFT in Section 4.3.

**Data Curation ($D_{EI}$).** The dataset $D_{EI}$ is constructed through a two-step filtering process applied to the candidate EI-augmented tuples $D_{cand}(Q)$ generated for each original question $Q$.

First, an **EI Consistency Filter** ensures that the probe and its teacher-generated reasoning do not detract from solving the original problem. Let $\text{Predict}(\mathcal{M}, \text{Prompt})$ denote the final answer predicted by a model $\mathcal{M}$ for a given prompt. A tuple $d_k$ passes this filter, *i.e.*, $\zeta_{EI}(d_k) = 1$, if the teacher model $\mathcal{T}$, when conditioned on the probe $Q_k^{\text{aug}}$, its reasoning $R_{T,k}^{\text{aug}}$, its answer $A_{T,k}^{\text{aug}}$, and then prompted with the original question $Q$, still correctly predicts the original answer $A$:

$$\zeta_{EI}(d_k) \Leftrightarrow \mathcal{T}([Q_k^{\text{aug}}||R_{T,k}^{\text{aug}}||A_{T,k}^{\text{aug}}||Q]) = A, \tag{1}$$

where $||$ indicates concatenation. Let $D'_{EI}(Q) = \{d_k \in D_{cand}(Q) \mid \zeta_{EI}(d_k) = 1\}$ be the set of EI-consistent probes for a specific original question $Q$, and $N'_Q = |D'_{EI}(Q)|$ be the count of such probes for that $Q$. Note that while $N'_Q \leq N$, the overall training process across many original questions $Q$ still exposes the student model to the full diversity of the $N$ EI rule types, as different rules will be applicable and pass filtering for different $Q$.

Second, we apply a **Rejective Filtering** step to remove original questions $Q$ (and all their associated probes in $D'_{EI}(Q)$) that are either "too easy" or "too hard" for a baseline student model $S_{base}$ (the student model before SFT) to learn effectively from their EI probes. Let $\lambda_{S_{base},j} = 1$ if $S_{base}$ correctly answers probe $Q_j^{\text{aug}} \in D'_{EI}(Q)$ with $A_{T,j}^{\text{aug}}$, and 0 otherwise. Let $\Lambda_{Q,S_{base}} = \sum_{j=1}^{N'_Q} \lambda_{S_{base},j}$ be the count of EI-consistent probes for $Q$ that $S_{base}$ answers correctly. An original question $Q$ (along with all its probes in $D'_{EI}(Q)$) is included in the final dataset $D_{EI}$ if:

$$\left( \neg(\Lambda_{Q,S_{base}} = N'_Q \wedge N'_Q > 0) \right) \wedge \left( \neg(\Lambda_{Q,S_{base}} \geq \tau_{hard} \wedge N'_Q > 0) \right), \tag{2}$$

where $\tau_{hard}$ is a predefined integer threshold ($\tau_{hard} \geq 1$). The first clause ensures that $S_{base}$ does not answer all $N'_Q$ EI-consistent probes correctly (i.e., at least one probe is challenging for $S_{base}$). The second clause ensures that $S_{base}$ answers at least $\tau_{hard}$ of the EI-consistent probes correctly. If $N'_Q = 0$, $Q$ is also discarded. The final dataset $D_{EI}$ comprises all tuples $(Q_k^{\text{aug}}, A_{T,k}^{\text{aug}}, R_{T,k}^{\text{aug}})$ from $D'_{EI}(Q)$ for all original questions $Q$ that pass this rejective filter. This $D_{EI}$ is used for the initial SFT, the source for teacher trajectories for $\mathcal{L}_{\text{SFT-aux}}$, and for sampling EI rule types in the RL stage.

### 3.3 STAGE 2: SUPERVISED FINE-TUNING FOR COLD START

The student model, with parameters $\theta$, is then fine-tuned on $D_{EI}$ for $P$ epochs. For each selected tuple $(Q_i^{\text{aug}}, A_{T,i}^{\text{aug}}, R_{T,i}^{\text{aug}}) \in D_{EI}$, the target output $T_i$ is the concatenation of the teacher's reasoning and answer $[R_{T,i}^{\text{aug}}||A_{T,i}^{\text{aug}}]$. The SFT objective is to maximize the likelihood of generating $T_i$ given $Q_i^{\text{aug}}$ by minimizing the negative log-likelihood loss:

$$\mathcal{L}_{SFT}(\theta) = - \sum_{(Q_i^{\text{aug}}, T_i) \in D_{EI}} \sum_{t=1}^{|T_i|} \log P(T_{i,t}|Q_i^{\text{aug}}, T_{i,<t}; \theta), \tag{3}$$

where $T_{i,t}$ is the $t$-th token of the target sequence $T_i$. This stage warms up the student model to the diverse reasoning challenges generated by EI, providing an initial alignment with teacher insights.

### 3.4 STAGE 3: REINFORCEMENT DISTILLATION VIA EXPLANATORY GRPO (EXGRPO).

Following SFT, the student model's policy $\pi_\theta$ (parameterized by $\theta$) is further refined using our ExGRPO algorithm. ExGRPO adapts Group Relative Policy Optimization (GRPO) to specifically enhance the distillation of reasoning by leveraging the structured interactions derived from EI.

### 3.4.1 INTERACTION PROTOCOL WITH RANDOMIZED EXPLANATORY PROBE SAMPLING

For an original question $Q$ in an ExGRPO training batch, a $k$-turn explanatory dialogue is constructed to probe and refine the student's reasoning. The construction and interaction unfold as follows:

**Probe Selection:** From the $N$ available Explanatory Inversion (EI) rule categories, $k$ distinct rule types are *randomly sampled without replacement* (where $k \leq N$, e.g., $k = 5$) to generate a sequence of $k$ augmented questions (probes $Q_1^{\text{aug}}, \ldots, Q_k^{\text{aug}}$).

**Sequential Interaction Formulation:** These $k$ probes are presented to the student one by one, and finally the original $Q$. For each turn $j \in \{1, \ldots, k\}$, the student receives the current probe $Q_j^{\text{aug}}$. The dialogue history, including all previous probe-response pairs $(Q_1^{\text{aug}}, R_{S,1}^{\text{aug}}, A_{S,1}^{\text{aug}}), \ldots, (Q_{j-1}^{\text{aug}}, R_{S,j-1}^{\text{aug}}, A_{S,j-1}^{\text{aug}})$, forms the context for this turn. Finally, the student generates its reasoning $(\hat{R}_S, \hat{A}_S)$ in response to $Q$, given the accumulated context. Note that each intermediate turn is supervised using an SFT loss $\mathcal{L}_{\text{SFT-aux}}$ defined in Equation (8). The final response $\hat{A}_S$ is used for computing rewards.

**Final Response Generation:** After completing all $k$ turns of interaction with the EI probes, the student model is prompted again with the original question $Q$ (now enriched by the preceding dialogue) to produce its final reasoning and answer for $Q$.

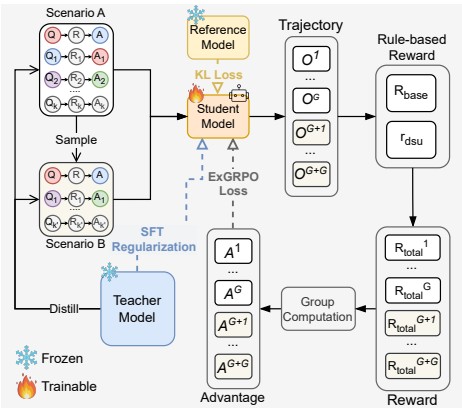

Figure 2: ExGRPO framework overview. The student model learns from multi-turn explanatory probe dialogues.

As shown in Figure 2, this entire sequence of $k$ probe interactions followed by the final answer to $Q$ constitutes a Full Dialogue (Scenario A). Randomly sampling a subset of EI probes for each dialogue instance offers several benefits: (1) It introduces diversity in training dialogues over epochs, encouraging robust and generalizable reasoning rather than overfitting to a fixed set or order of $N$ probes. (2) It enhances computational efficiency during RL compared to processing all $N$ probes in every $k$-turn dialogue if $k$ were always $N$. (3) It allows for focused yet varied probing in each dialogue, preventing potential cognitive overwhelm from too many simultaneous challenges.

For the purpose of computing the **Dialogue Structure Utility Bonus** ($r_{\text{dsu}}$), contrastive trajectories under a Partial Dialogue (Scenario B) are also generated for a subset of training instances. Scenario B is constructed similarly but involves interaction with only $k'$ probes sampled from the $k$ probes ($k' < k$, e.g., $k'/2$ turns) before the student answers the original question $Q$. A group of $G$ trajectories is generated for each scenario (Scenario A and Scenario B) per input $Q$.

### 3.4.2 RULE-BASED REWARD DESIGN FOR ExGRPO

Each of the $G$ generated trajectories receives a rule-based reward. Let $g$ be the trajectory index.

**Outcome Reward** ($R_{\text{outcome}}$): The primary reward, assessing the correctness of the student's final answer to $Q$ for trajectory $g$.

$$R_{\text{outcome}}^{(g)}(Q, A) = \begin{cases} 1, & \text{if student's final answer in trajectory } g \text{ to } Q \text{ matches ground-truth } A, \\ 0, & \text{otherwise.} \end{cases} \quad (4)$$

This serves as the base reward: $R_{\text{base}}^{(g)} = R_{\text{outcome}}^{(g)}$.

**Dialogue Structure Utility Bonus** ($r_{\text{dsu}}$): This bonus rewards the student if engaging in the full $k$-turn probing dialogue (Scenario A) leads to better overall outcomes than a partial $k'$-turn dialogue (Scenario B). Let $P_{\text{full}}$ be average $R_{\text{base}}$ for Scenario A trajectories, and $P_{\text{partial}}$ for Scenario B:

$$r_{\text{dsu}} = \begin{cases} \delta, & \text{if } P_{\text{full}} > \nu \cdot P_{\text{partial}}, \\ 0, & \text{otherwise,} \end{cases} \quad (5)$$

where $\delta > 0$ is the bonus value and $\nu \geq 1.0$ is a performance margin. This bonus encourages effective learning from the entire sequence of explanatory probes.

**Total Augmented Rule-Based Reward** ($R_{\text{total}}$): For trajectory $g$ from Scenario A:

$$R_{\text{total}}^{(g)} = \begin{cases} R_{\text{base}}^{(g)} + r_{\text{dsu}}, & \text{if } R_{\text{outcome}}^{(g)} = 1 \text{ (from Scenario A) and Equation (5) yields } \delta, \\ R_{\text{base}}^{(g)}, & \text{otherwise.} \end{cases} \quad (6)$$

### 3.4.3 ADVANTAGE COMPUTATION AND POLICY UPDATE

Advantages $U^{(g)}$ are computed by normalizing $\{R_{\text{total}}^{(m)}\}_{m=1}^{G}$ within the group. The policy $\pi_\theta$ is updated using the ExGRPO objective, $\mathcal{L}_{\text{ExGRPO}}(\theta)$:

$$\mathcal{L}_{\text{ExGRPO}}(\theta) = \mathbb{E}_{\text{traj}_g \sim \pi_{\theta_{\text{old}}}} \left[ \sum_{g=1}^{G} \min \left( \rho^{(g)}(\theta) U^{(g)}, \text{clip}(\rho^{(g)}(\theta), 1 - \epsilon, 1 + \epsilon) U^{(g)} \right) \right] - \beta \mathbb{D}_{\text{KL}}(\pi_\theta || \pi_{\text{ref}}), \quad (7)$$

where $\pi_{\theta_{\text{old}}}$ is the policy before update, $\rho^{(g)}(\theta)$ is the probability ratio, $\epsilon$ is the clipping hyperparameter, $\pi_{\text{ref}}$ is a reference policy, and $\beta$ is the KL coefficient. We set the model after stage 1 as the reference.

### 3.4.4 IMITATION-BASED POLICY REGULARIZATION:

To guide student reasoning during the EI turns, an auxiliary SFT loss, $\mathcal{L}_{\text{SFT-aux}}$, encourages imitation of teacher responses $R_{T,j}^{\text{aug}}$:

$$\mathcal{L}_{\text{SFT-aux}} = -\sum_{j=1}^{k} \log \pi_\theta(R_{T,j}^{\text{aug}} | Q_j^{\text{aug}}, \text{context}). \quad (8)$$

This loss is combined with or alternated with $\mathcal{L}_{\text{ExGRPO}}$. In practice, we add more regularization losses for training stability. See details in Appendix D.

**Theorem 3.1.** *Let $\pi_k$ and $\pi_{k'}$ be student policies trained with full ($k$-turn) and partial ($k' < k$) explanatory probe sequences, respectively. If the utility bonus $r_{dsu}$ is applied only when:*

$$\mathbb{E}_{\pi_k}[R_{outcome}(Q, A)] > \nu \cdot \mathbb{E}_{\pi_{k'}}[R_{outcome}(Q, A)] \quad \text{for some } \nu \geq 1. \quad (9)$$

*Then the ExGRPO policy update with clipped importance sampling ensures:*

$$\mathbb{E}_{\pi_{new}}[R_{outcome}(Q, A)] \geq \mathbb{E}_{\pi_k}[R_{outcome}(Q, A)], \quad (10)$$

*with strict inequality if $r_{dsu} > 0$ for any training instance.*

The proof is provided in Appendix J. With the derived theorem, we guarantee that learning from coherent and complete explanatory dialogues leads to performance improvements.

### 3.5 INFERENCE PROCEDURE

During inference, the trained student model $\pi_\theta$ generates reasoning and the final answer for an input $Q$ in a single pass. The multi-turn probing dialogues, random rule sampling, and reward computations (including $r_{\text{dsu}}$ logic) are training-time mechanisms for refinement.

## 4 EXPERIMENTS

### 4.1 EXPERIMENTAL SETUP

**Datasets.** For training, we follow (Chen et al., 2025a) and utilize a diverse set of source datasets covering various reasoning tasks: *Commonsense reasoning:* StrategyQA (SQA (Geva et al., 2021)), CommonsenseQA (CSQA (Talmor et al., 2018)), ARC-challenge (ARC (Clark et al., 2018)). *Math reasoning:* MATH (Hendrycks et al., 2021), GSM8K (Cobbe et al., 2021). *Tabular data reasoning:* TabMWP (Lu et al., 2022). *Natural Language Inference:* ANLI (Nie et al., 2019). *Logical Reasoning:* Date Understanding (Srivastava et al., 2022). Our Explanatory Inversion (EI) augmentation process employs Gemini-1.5-Pro as the teacher model to generate responses to $N = 10$ categories of EI probes (detailed in Appendix E). Statistics of the augmented dataset are provided in Appendix G.

**Models.** Our main experiments employ a *weaker* Gemma-7b-it and a *stronger* Qwen2.5-7b-instruct as student models. Following (Chen et al., 2025a), we compare our approach against these baselines. *(1) Zero-shot:* We compare with the student's zero-shot performance (Kojima et al., 2022) as a reference. *(2) Knowledge distillation:* We compare with Symbolic Knowledge Distillation (SKD (Li et al., 2023; West et al., 2021)), Distilling Step-by-Step (Hsieh et al., 2023), and On-Policy Distillation (Agarwal

Table 1: Main results comparing our `ExGRPO` against zero-shot performance and various knowledge distillation and data augmentation baselines across eight held-in reasoning datasets for Qwen2.5-7B-Instruct and Gemma-7B-it student models. Teacher model performance is also shown for reference. `ExGRPO` shows consistent improvements. Scores are reported in percentage (%). * indicates the score is quoted from RevThink (Chen et al., 2025a). The contribution of each rule is studied in Appendix F.

| Methods | SQA | CSQA | ARC-c | MATH | GSM8K | TabMWP | ANLI | Date | Avg. |
|---|---|---|---|---|---|---|---|---|---|
| *Gemini-1.5-Pro (Teacher Model)* | | | | | | | | | |
| Zero-shot (Kojima et al., 2022) | 78.60 | 77.90 | 92.00 | 77.20 | 94.50 | 95.10 | 71.00 | 81.50 | 83.48 |
| Zero-shot-EI [Ours] | $78.55_{\uparrow 0.05}$ | $78.78_{\uparrow 0.88}$ | $96.82_{\uparrow 4.82}$ | $80.59_{\uparrow 3.39}$ | $93.51_{\downarrow 0.99}$ | $97.61_{\uparrow 2.51}$ | $74.02_{\uparrow 3.02}$ | $85.75_{\uparrow 4.25}$ | $85.70_{\uparrow 2.22}$ |
| *Qwen2.5-7B-Instruct (Student Model)* | | | | | | | | | |
| Zero-shot (Kojima et al., 2022) | 72.93 | 71.33 | 85.36 | 74.40 | 90.90 | 93.88 | 61.67 | 73.43 | 77.99 |
| SKD (Li et al., 2023; West et al., 2022) | $73.51_{\uparrow 0.58}$ | $73.12_{\uparrow 1.79}$ | $87.26_{\uparrow 1.90}$ | $74.88_{\uparrow 0.48}$ | $86.20_{\downarrow 4.70}$ | $94.17_{\uparrow 0.29}$ | $62.73_{\uparrow 1.06}$ | $75.10_{\uparrow 1.67}$ | $78.01_{\uparrow 0.02}$ |
| Distill Step-by-Step (Hsieh et al., 2023) | $74.12_{\uparrow 1.19}$ | $74.33_{\uparrow 3.00}$ | $88.40_{\uparrow 3.04}$ | $75.32_{\uparrow 0.92}$ | $86.85_{\downarrow 4.05}$ | $94.76_{\uparrow 0.88}$ | $63.58_{\uparrow 1.91}$ | $76.42_{\uparrow 2.99}$ | $78.65_{\uparrow 0.66}$ |
| Rephrase Question (Yu et al., 2024) | $75.26_{\uparrow 2.33}$ | $75.18_{\uparrow 3.85}$ | $89.05_{\uparrow 3.69}$ | $75.64_{\uparrow 1.24}$ | $87.21_{\downarrow 3.69}$ | $95.12_{\uparrow 1.24}$ | $63.95_{\uparrow 2.28}$ | $77.80_{\uparrow 4.37}$ | $79.19_{\uparrow 1.20}$ |
| Question Aug (Li et al., 2024) | $75.70_{\uparrow 2.77}$ | $76.40_{\uparrow 5.07}$ | $89.64_{\uparrow 4.28}$ | $75.97_{\uparrow 1.57}$ | $87.60_{\downarrow 3.30}$ | $95.67_{\uparrow 1.79}$ | $64.45_{\uparrow 2.78}$ | $78.21_{\uparrow 4.78}$ | $79.81_{\uparrow 1.82}$ |
| Answer Aug (Yu et al., 2024) | $76.20_{\uparrow 3.27}$ | $77.36_{\uparrow 6.03}$ | $90.10_{\uparrow 4.74}$ | $76.18_{\uparrow 1.78}$ | $87.92_{\downarrow 2.98}$ | $96.12_{\uparrow 2.24}$ | $64.83_{\uparrow 3.16}$ | $78.74_{\uparrow 5.31}$ | $80.34_{\uparrow 2.35}$ |
| RevThink (Chen et al., 2025a) | $77.02_{\uparrow 4.09}$ | $78.45_{\uparrow 7.12}$ | $90.55_{\uparrow 5.19}$ | $76.40_{\uparrow 2.00}$ | $88.00_{\downarrow 2.90}$ | $96.73_{\uparrow 2.85}$ | $65.10_{\uparrow 3.43}$ | $79.01_{\uparrow 5.58}$ | $80.89_{\uparrow 2.90}$ |
| Divide-or-Conquer (Wu et al., 2024) | $77.15_{\uparrow 4.22}$ | $77.42_{\uparrow 6.09}$ | $90.12_{\uparrow 4.76}$ | $76.05_{\uparrow 1.65}$ | $88.43_{\downarrow 2.47}$ | $95.62_{\uparrow 1.74}$ | $63.89_{\uparrow 2.22}$ | $74.92_{\uparrow 1.49}$ | $80.45_{\uparrow 2.46}$ |
| On-Policy Distillation (Agarwal et al., 2024) | $78.12_{\uparrow 5.19}$ | $79.05_{\uparrow 7.72}$ | $91.08_{\uparrow 5.72}$ | $76.24_{\uparrow 1.84}$ | $89.55_{\downarrow 1.35}$ | $96.15_{\uparrow 2.27}$ | $64.18_{\uparrow 2.51}$ | $78.43_{\uparrow 5.00}$ | $81.60_{\uparrow 3.61}$ |
| ExGRPO (Ours) | $\mathbf{79.04}_{\uparrow 6.11}$ | $80.85_{\uparrow 9.52}$ | $91.45_{\uparrow 6.09}$ | $76.58_{\uparrow 2.18}$ | $91.46_{\uparrow 0.56}$ | $97.10_{\uparrow 3.22}$ | $64.00_{\uparrow 2.33}$ | $79.80_{\uparrow 6.37}$ | $82.54_{\uparrow 4.55}$ |
| ExGRPO (On-Policy Adapt.) | $\mathbf{79.62}_{\uparrow 6.69}$ | $\mathbf{81.45}_{\uparrow 10.12}$ | $\mathbf{91.78}_{\uparrow 6.42}$ | $\mathbf{77.12}_{\uparrow 2.72}$ | $\mathbf{91.85}_{\uparrow 0.95}$ | $\mathbf{97.43}_{\uparrow 3.55}$ | $64.42_{\uparrow 2.75}$ | $\mathbf{81.13}_{\uparrow 7.70}$ | $\mathbf{83.10}_{\uparrow 5.11}$ |
| *Gemma-7B-it (Student Model)* | | | | | | | | | |
| Zero-shot (Kojima et al., 2022) | 56.33* | 66.26* | 68.34* | 8.58* | 41.09* | 55.67* | 37.92* | 40.24* | 46.80* |
| SKD (Li et al., 2023; West et al., 2022) | $56.77^*_{\uparrow 0.44}$ | $72.48^*_{\uparrow 6.22}$ | $73.29^*_{\uparrow 4.95}$ | $16.86^*_{\uparrow 8.28}$ | $52.24^*_{\uparrow 11.15}$ | $60.52^*_{\uparrow 4.85}$ | $45.42^*_{\uparrow 7.50}$ | $59.62^*_{\uparrow 19.38}$ | $54.65^*_{\uparrow 7.85}$ |
| Distill Step-by-Step (Hsieh et al., 2023) | $56.77^*_{\uparrow 0.44}$ | $73.01^*_{\uparrow 6.75}$ | $72.92^*_{\uparrow 4.58}$ | $16.04^*_{\uparrow 7.46}$ | $51.88^*_{\uparrow 10.79}$ | $62.11^*_{\uparrow 6.44}$ | $44.23^*_{\uparrow 6.31}$ | $60.91^*_{\uparrow 20.67}$ | $54.73^*_{\uparrow 7.93}$ |
| Rephrase Question (Yu et al., 2024) | $54.15^*_{\downarrow 2.18}$ | $70.22^*_{\uparrow 3.96}$ | $72.37^*_{\uparrow 4.03}$ | $16.96^*_{\uparrow 8.38}$ | $53.07^*_{\uparrow 11.98}$ | $57.62^*_{\uparrow 1.95}$ | $43.07^*_{\uparrow 5.15}$ | $57.99^*_{\uparrow 17.75}$ | $53.18^*_{\uparrow 6.38}$ |
| Question Aug (Li et al., 2024) | $55.10^*_{\downarrow 1.23}$ | $68.11^*_{\uparrow 1.85}$ | $72.74^*_{\uparrow 4.40}$ | $17.76^*_{\uparrow 9.18}$ | $56.38^*_{\uparrow 15.29}$ | $63.16^*_{\uparrow 7.49}$ | $41.22^*_{\uparrow 3.30}$ | $59.83^*_{\uparrow 19.59}$ | $54.29^*_{\uparrow 7.49}$ |
| Answer Aug (Yu et al., 2024) | $57.21^*_{\uparrow 0.88}$ | $73.01^*_{\uparrow 6.76}$ | $73.92^*_{\uparrow 5.58}$ | $18.92^*_{\uparrow 10.34}$ | $57.37^*_{\uparrow 16.28}$ | $65.93^*_{\uparrow 10.26}$ | $42.72^*_{\uparrow 4.80}$ | $64.14^*_{\uparrow 23.90}$ | $56.65^*_{\uparrow 9.85}$ |
| RevThink (Chen et al., 2025a) | $64.19^*_{\uparrow 7.86}$ | $74.53^*_{\uparrow 8.27}$ | $75.09^*_{\uparrow 6.75}$ | $19.96^*_{\uparrow 11.38}$ | $57.21^*_{\uparrow 16.12}$ | $84.71^*_{\uparrow 29.04}$ | $47.36^*_{\uparrow 9.44}$ | $66.27^*_{\uparrow 26.03}$ | $61.17^*_{\uparrow 14.37}$ |
| Divide-or-Conquer (Wu et al., 2024) | $60.12_{\uparrow 3.79}$ | $70.15_{\uparrow 3.89}$ | $74.23_{\uparrow 5.89}$ | $18.15_{\uparrow 9.57}$ | $56.12_{\uparrow 15.03}$ | $75.18_{\uparrow 19.51}$ | $47.45_{\uparrow 9.53}$ | $69.16_{\uparrow 28.92}$ | $58.82_{\uparrow 12.02}$ |
| On-Policy Distillation (Agarwal et al., 2024) | $66.23_{\uparrow 9.90}$ | $74.12_{\uparrow 7.86}$ | $78.15_{\uparrow 9.81}$ | $22.34_{\uparrow 13.76}$ | $60.45_{\uparrow 19.36}$ | $86.21_{\uparrow 30.54}$ | $52.18_{\uparrow 14.26}$ | $83.52_{\uparrow 43.28}$ | $65.40_{\uparrow 18.60}$ |
| ExGRPO (Ours) | $\mathbf{69.43}_{\uparrow 13.10}$ | $76.82_{\uparrow 10.56}$ | $79.94_{\uparrow 11.60}$ | $25.82_{\uparrow 17.24}$ | $65.27_{\uparrow 24.18}$ | $90.55_{\uparrow 34.88}$ | $55.75_{\uparrow 17.83}$ | $\mathbf{73.96}_{\uparrow 33.72}$ | $67.19_{\uparrow 20.39}$ |
| ExGRPO (On-Policy Adapt.) | $\mathbf{70.65}_{\uparrow 14.32}$ | $\mathbf{77.42}_{\uparrow 11.16}$ | $\mathbf{80.67}_{\uparrow 12.33}$ | $\mathbf{26.45}_{\uparrow 17.87}$ | $\mathbf{66.12}_{\uparrow 25.03}$ | $\mathbf{91.23}_{\uparrow 35.56}$ | $\mathbf{56.34}_{\uparrow 18.42}$ | $75.52_{\uparrow 35.28}$ | $\mathbf{68.05}_{\uparrow 21.25}$ |

et al., 2024). *(3) Data augmentation:* This set of baselines uses various methods to augment the dataset while applying the same next-token prediction objective. We compare with: Question Rephrasing (Yu et al., 2023), Question Augmentation (Li et al., 2024), Answer Augmentation (Yu et al., 2023), Divide-or-Conquer (Wu et al., 2024), and RevThink (Chen et al., 2025a). Their detailed descriptions are given in Appendix H.

**Evaluation Metrics & Implementation Details.** The primary metric for evaluation across all tasks is accuracy. Specific values for all hyperparameters (*e.g.*, learning rate) are detailed in Appendix I.

## 4.2 MAIN RESULTS: OVERALL PERFORMANCE

Table 1 presents the main experimental results Our `ExGRPO` method demonstrates a clear and consistent advantage, achieving an average accuracy of **82.54%** for Qwen and **67.19%** for Gemma. This surpasses all baselines. `ExGRPO` outperforms standard knowledge distillation techniques such as Distill Step-by-Step, as well as strong data augmentation strategies such as RevThink. The impact of `ExGRPO` is particularly notable when contrasted across student models of **varying initial strengths**. For the stronger Qwen model, `ExGRPO` yields a significant average improvement of **+4.55%** over its zero-shot performance. For the initially weaker Gemma model, `ExGRPO` delivers a remarkable **+20.39%** average improvement, showing its effectiveness in enhancing reasoning potential, especially for less capable base models. Notably, on the GSM8K dataset, where the strong Qwen student model already has a strong performance, `ExGRPO` is the only method that yields positive transfer. This **consistent pattern** of strong improvement across both student models and individual datasets highlights the robustness and broad applicability of our approach. Interestingly, our EI-based data augmentation also improved the Gemini-1.5-Pro teacher's zero-shot performance by **+2.22%** (85.71% "Zero-shot-EI" vs. 83.48% zero-shot), suggesting EI can beneficially structure reasoning tasks even for powerful models (We further show the improvement of each EI logic in Appendix E). The best student performance (Qwen with `ExGRPO` at **82.54%**) approaches the teacher's zero-shot capability, demonstrating effective knowledge transfer. In summary, `ExGRPO`, by probing and refining reasoning pathways via EI and our novel RL strategy, achieves state-of-the-art results among evaluated methods, significantly enhancing distilled LLM reasoning across different capabilities.

## 4.3 ABLATION STUDY

Table 3 presents our ablation study on the impact of SFT warm-up and key RL components within `ExGRPO`, using the same EI-augmented datasets for all variants. The results show several critical factors. Firstly, different from post-training, most of the time **SFT is useful for distillation**; RL from a cold start with only $R_{\text{base}}$ results in catastrophic performance degradation for both Qwen (avg.

Table 3: Ablation study on the impact of SFT warm-up training and RL components for ExGRPO. We evaluate performance across eight reasoning datasets. All model variants are trained using the same augmented datasets with EI probes. RL without $r_{\mathrm{dsu}}$ treats each EI probe as an independent training sample without grouping across reasoning paths.

| Model Configuration | SQA | CSQA | ARC-c | MATH | GSM8K | TabMWP | ANLI | Date | Avg. |
|---|---|---|---|---|---|---|---|---|---|
| *Qwen2.5-7B-Instruct (Student Model)* | | | | | | | | | |
| Zero-shot | 72.93 | 71.33 | 85.36 | 74.40 | 90.90 | 93.88 | 61.67 | 73.43 | 77.99 |
| SFT (1 epoch only) | 73.42$_{\uparrow0.49}$ | 72.10$_{\uparrow0.77}$ | 85.90$_{\uparrow0.54}$ | 74.80$_{\uparrow0.40}$ | 86.55$_{\downarrow4.35}$ | 94.12$_{\uparrow0.24}$ | 62.10$_{\uparrow0.43}$ | 74.02$_{\uparrow0.59}$ | 77.88$_{\downarrow0.11}$ |
| SFT (3 epochs only) | 76.40$_{\uparrow3.47}$ | 74.50$_{\uparrow3.17}$ | 89.30$_{\uparrow3.94}$ | 74.65$_{\uparrow0.25}$ | 86.60$_{\downarrow4.30}$ | 94.25$_{\uparrow0.37}$ | 62.75$_{\uparrow1.08}$ | 76.90$_{\uparrow3.47}$ | 79.17$_{\uparrow1.18}$ |
| RL (cold start, $R_{\mathrm{base}}$) | 10.20$_{\downarrow62.73}$ | 15.80$_{\downarrow55.53}$ | 18.60$_{\downarrow66.76}$ | 5.10$_{\downarrow69.30}$ | 20.90$_{\downarrow70.00}$ | 22.30$_{\downarrow71.58}$ | 13.40$_{\downarrow48.27}$ | 18.60$_{\downarrow54.83}$ | 15.99$_{\downarrow62.00}$ |
| RL (cold start, $R_{\mathrm{base}}$) + $\mathcal{L}_{\mathrm{SFT\text{-}aux}}$ | 63.10$_{\downarrow9.83}$ | 62.45$_{\downarrow8.88}$ | 79.40$_{\downarrow5.96}$ | 66.55$_{\downarrow7.85}$ | 85.90$_{\downarrow5.00}$ | 88.95$_{\downarrow4.93}$ | 54.80$_{\downarrow6.87}$ | 67.10$_{\downarrow6.33}$ | 71.53$_{\downarrow6.46}$ |
| SFT (1 ep) + RL ($R_{\mathrm{base}}$) | 71.90$_{\uparrow1.03}$ | 70.20$_{\uparrow1.13}$ | 84.10$_{\uparrow1.26}$ | 73.20$_{\uparrow1.20}$ | 89.50$_{\downarrow1.40}$ | 93.60$_{\downarrow0.28}$ | 60.80$_{\uparrow0.87}$ | 72.20$_{\uparrow1.23}$ | 76.69$_{\uparrow1.30}$ |
| SFT (3 ep) + RL ($R_{\mathrm{base}}$) | 76.21$_{\uparrow3.28}$ | 78.60$_{\uparrow7.27}$ | 89.22$_{\uparrow3.86}$ | 74.10$_{\downarrow0.30}$ | 90.40$_{\downarrow0.50}$ | 96.40$_{\uparrow2.52}$ | 63.20$_{\uparrow1.53}$ | 76.80$_{\uparrow3.37}$ | 80.13$_{\uparrow2.14}$ |
| SFT (3 ep) + RL ($R_{\mathrm{base}}$) + $\mathcal{L}_{\mathrm{SFT\text{-}aux}}$ | 77.30$_{\uparrow4.37}$ | 79.50$_{\uparrow8.17}$ | 90.10$_{\uparrow4.74}$ | 75.40$_{\uparrow1.00}$ | 90.80$_{\uparrow0.10}$ | 96.85$_{\uparrow2.97}$ | 63.75$_{\uparrow2.08}$ | 78.10$_{\uparrow4.67}$ | 81.23$_{\uparrow3.24}$ |
| SFT (3 ep) + ExGRPO ($R_{\mathrm{base}} + r_{\mathrm{dsu}}$) + $\mathcal{L}_{\mathrm{SFT\text{-}aux}}$ | **79.04**$_{\uparrow6.11}$ | **80.85**$_{\uparrow9.52}$ | **91.45**$_{\uparrow6.09}$ | **76.58**$_{\uparrow2.18}$ | **91.46**$_{\uparrow0.56}$ | **97.10**$_{\uparrow3.22}$ | **64.00**$_{\uparrow2.33}$ | **79.80**$_{\uparrow6.37}$ | **82.54**$_{\uparrow4.55}$ |
| *Gemma-7B-it (Student Model)* | | | | | | | | | |
| Zero-shot | 56.33 | 66.26 | 68.34 | 8.58 | 41.09 | 55.67 | 37.92 | 40.24 | 46.80 |
| SFT (1 epoch only) | 53.90$_{\downarrow-2.43}$ | 64.13$_{\downarrow-2.13}$ | 66.06$_{\downarrow-2.28}$ | 6.36$_{\downarrow-2.22}$ | 37.35$_{\downarrow-3.74}$ | 53.33$_{\downarrow-2.34}$ | 35.45$_{\downarrow-2.47}$ | 38.20$_{\downarrow-2.04}$ | 44.35$_{\downarrow-2.45}$ |
| SFT (3 epochs only) | 63.93$_{\uparrow+7.60}$ | 70.83$_{\uparrow+4.57}$ | 73.83$_{\uparrow+5.49}$ | 15.89$_{\uparrow7.31}$ | 37.74$_{\downarrow-3.35}$ | 63.27$_{\uparrow+7.60}$ | 40.64$_{\uparrow+10.72}$ | 57.30$_{\uparrow+17.06}$ | 52.93$_{\uparrow+7.3}$ |
| RL (cold start, $R_{\mathrm{base}}$) | 9.10$_{\downarrow47.23}$ | 12.40$_{\downarrow53.86}$ | 14.20$_{\downarrow54.14}$ | 3.80$_{\downarrow4.78}$ | 13.75$_{\downarrow27.34}$ | 18.20$_{\downarrow37.47}$ | 11.90$_{\downarrow26.02}$ | 15.33$_{\downarrow24.91}$ | 12.34$_{\downarrow34.46}$ |
| RL (cold start, $R_{\mathrm{base}}$) + $\mathcal{L}_{\mathrm{SFT\text{-}aux}}$ | 50.10$_{\downarrow6.23}$ | 62.40$_{\downarrow3.86}$ | 65.30$_{\downarrow3.04}$ | 7.20$_{\downarrow1.38}$ | 38.60$_{\downarrow2.49}$ | 52.90$_{\downarrow2.77}$ | 35.60$_{\downarrow2.32}$ | 38.80$_{\downarrow1.44}$ | 43.49$_{\downarrow3.31}$ |
| SFT (1 ep) + RL ($R_{\mathrm{base}}$) | 54.80$_{\downarrow1.53}$ | 64.00$_{\downarrow2.26}$ | 67.21$_{\downarrow1.13}$ | 7.81$_{\downarrow0.77}$ | 39.60$_{\downarrow1.49}$ | 55.44$_{\downarrow0.23}$ | 36.10$_{\downarrow1.82}$ | 39.30$_{\downarrow0.94}$ | 45.79$_{\downarrow1.01}$ |
| SFT (3 ep) + RL ($R_{\mathrm{base}}$) | 66.11$_{\uparrow9.78}$ | 74.12$_{\uparrow7.86}$ | 76.43$_{\uparrow8.09}$ | 22.20$_{\uparrow13.62}$ | 40.70$_{\downarrow0.39}$ | 86.30$_{\uparrow30.63}$ | 52.60$_{\uparrow14.68}$ | 69.01$_{\uparrow28.77}$ | 58.94$_{\uparrow12.14}$ |
| SFT (3 ep) + RL ($R_{\mathrm{base}}$) + $\mathcal{L}_{\mathrm{SFT\text{-}aux}}$ | 67.70$_{\uparrow11.37}$ | 75.01$_{\uparrow8.75}$ | 77.80$_{\uparrow9.46}$ | 23.95$_{\uparrow15.37}$ | 43.00$_{\uparrow1.91}$ | 88.00$_{\uparrow32.33}$ | 53.40$_{\uparrow15.48}$ | 71.50$_{\uparrow31.26}$ | 61.80$_{\uparrow14.99}$ |
| SFT (3 ep) + ExGRPO ($R_{\mathrm{base}} + r_{\mathrm{dsu}}$) + $\mathcal{L}_{\mathrm{SFT\text{-}aux}}$ | **69.43**$_{\uparrow13.10}$ | **76.82**$_{\uparrow10.56}$ | **79.94**$_{\uparrow11.60}$ | **25.82**$_{\uparrow17.24}$ | **65.27**$_{\uparrow24.18}$ | **90.55**$_{\uparrow34.88}$ | **55.75**$_{\uparrow17.83}$ | **73.96**$_{\uparrow33.72}$ | **67.19**$_{\uparrow20.39}$ |

Figure 3: RL training curves with Gemma as student. Evolution of key reward components during ExGRPO training. Left: $R_{base}$. Right: $r_{dsu}$, scaled for visualization.

-62.00% drop from zero-shot) and Gemma (avg. **-34.46%** drop from zero-shot). However, **vanilla or insufficient SFT does not always improve the student model**, especially when the model is strong, as shown for the GSM8K dataset using Qwen2.5-7b as the student. Secondly, $\mathcal{L}_{\mathrm{SFT\text{-}aux}}$, **the imitation-based policy regularization, proves highly beneficial**. Adding $\mathcal{L}_{\mathrm{SFT\text{-}aux}}$ to cold-start RL recovers significant performance, bringing avg. **+9.53%** gain to Qwen and **+31.15%** gain to Gemma-7B. When combined with a 3-epoch SFT warm-up, $\mathcal{L}_{\mathrm{SFT\text{-}aux}}$ further lifts students' performance. Finally, the introduction of $r_{\mathbf{dsu}}$ **provides the crucial boost**, elevating Qwen2.5-7B to 82.54% (an additional **+1.31%**) and Gemma-7B to 67.19% (an additional **+5.39%**). This demonstrates that while SFT warm-up and $\mathcal{L}_{\mathrm{SFT\text{-}aux}}$ are vital for stabilizing RL and guiding intermediate reasoning, the $r_{\mathrm{dsu}}$ component uniquely incentivizes coherent processing of the entire explanatory dialogue. A post-hoc analysis of each individual EI rule is given in Appendix F.

## 4.4 Training Dynamics: Reward Evolution

Figure 3 depicts the training dynamics of key reward components within our ExGRPO framework. The average base outcome reward ($R_{\mathrm{base}}$), reflecting final answer correctness, steadily increases from approximately **0.6** to stabilize near **0.8** after around **0.3 epochs**, indicating the student model's improving ability to solve the primary task. Concurrently, $r_{\mathrm{dsu}}$, which rewards effective use of the multi-turn explanatory dialogue, exhibits a similar upward trend and stabilization. This synchronized improvement suggests that the ExGRPO policy successfully learns to achieve correct outcomes, potentially by leveraging the structured interactions with explanatory probes to foster deeper reasoning.

Table 2: OOD generalization on four held-out datasets. ExGRPO significantly improves generalization across both Qwen2.5-7B and Gemma-7B.

| Method | BoolQ | Openbook | e-SNLI | GSM8K-Rev | Avg. |
|---|---|---|---|---|---|
| *Qwen2.5-7B-Instruct (student)* | | | | | |
| Zero-shot | 71.25 | 77.80 | 69.33 | 75.10 | 73.87 |
| SKD | 74.62$_{\uparrow3.37}$ | 79.80$_{\uparrow2.00}$ | 71.20$_{\uparrow1.87}$ | 78.45$_{\uparrow3.35}$ | 76.52$_{\uparrow2.65}$ |
| AnsAug | 75.53$_{\uparrow4.28}$ | 80.30$_{\uparrow2.50}$ | 70.80$_{\uparrow1.47}$ | 79.62$_{\uparrow4.52}$ | 76.56$_{\uparrow2.69}$ |
| RevThink | 77.10$_{\uparrow5.85}$ | 82.85$_{\uparrow5.05}$ | 73.91$_{\uparrow4.58}$ | 82.40$_{\uparrow7.30}$ | 79.06$_{\uparrow5.19}$ |
| ExGRPO | **80.30**$_{\uparrow9.05}$ | **86.41**$_{\uparrow8.61}$ | **78.44**$_{\uparrow9.11}$ | **86.22**$_{\uparrow11.12}$ | **82.34**$_{\uparrow8.47}$ |
| *Gemma-7B-it (student)* | | | | | |
| Zero-shot | 53.18 | 70.20 | 52.27 | 17.37 | 48.76 |
| SKD | 58.65$_{\uparrow5.47}$ | 73.12$_{\uparrow2.92}$ | 54.40$_{\uparrow2.13}$ | 26.05$_{\uparrow8.68}$ | 53.56$_{\uparrow4.80}$ |
| AnsAug | 59.42$_{\uparrow6.24}$ | 73.80$_{\uparrow3.60}$ | 53.75$_{\uparrow1.48}$ | 26.90$_{\uparrow9.53}$ | 53.97$_{\uparrow5.21}$ |
| RevThink | 61.85$_{\uparrow8.67}$ | 77.20$_{\uparrow7.00}$ | 58.30$_{\uparrow6.03}$ | 30.12$_{\uparrow12.75}$ | 56.87$_{\uparrow8.11}$ |
| ExGRPO | **66.23**$_{\uparrow13.05}$ | **80.70**$_{\uparrow10.50}$ | **63.21**$_{\uparrow10.94}$ | **34.88**$_{\uparrow17.51}$ | **61.76**$_{\uparrow13.00}$ |

by leveraging the structured interactions with explanatory probes to foster deeper reasoning.

## 4.5 Generalization to Out-of-Distribution Datasets

We further evaluate ExGRPO's ability to generalize to datasets unseen during distillation following Chen et al. (2025a), a critical indicator of robust reasoning. Table 2 compares OOD performance

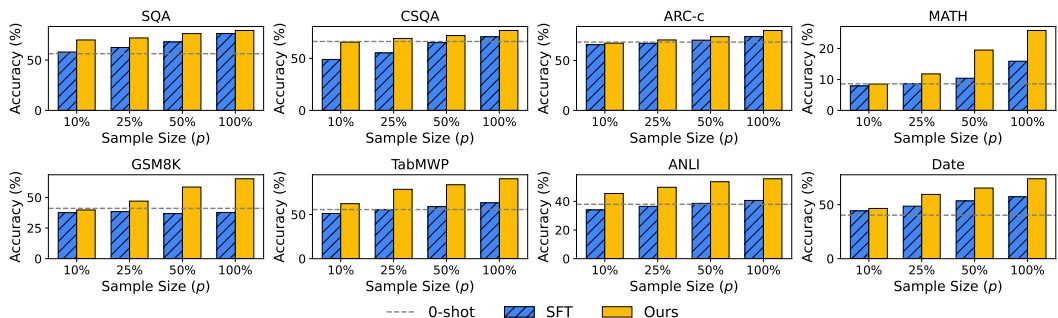

Figure 4: Sample efficiency comparison on eight datasets. Our `ExGRPO` method achieves higher accuracy than standard SFT across all training data fractions ($p \in \{0.1, 0.25, 0.5, 1.0\}$), often surpassing SFT trained on the full dataset with only $10 - 25\%$ of the data.

against baselines on four held-out datasets: BoolQ (Clark et al., 2019) (trained on StrategyQA), OpenbookQA (Mihaylov et al., 2018) (trained on ARC-c), e-SNLI (Camburu et al., 2018) (trained on ANLI), and GSM8K-Reversal (Guo et al., 2024) (trained on GSM8K).

Our `ExGRPO` method demonstrates superior generalization. For the Gemma student model, `ExGRPO` outperforms RevThink by **+4.89%** on average, with consistent gains across all four OOD tasks Notably, for the stronger Qwen student, `ExGRPO` surpasses RevThink by a significant margin of **+3.28%** on average. This includes strong individual gains such as **+3.82%** on GSM8K-Reversal. These results suggest that `ExGRPO`'s approach of probing and refining reasoning pathways improves the generalizability of learned reasoning skills to unseen domains.

## 4.6 EFFICIENCY STUDY

**Data Sample Efficiency.** Figure 4 illustrates the data sample efficiency of our `ExGRPO` compared to SFT and the student model's zero-shot performance, evaluated across eight reasoning datasets using varying percentages ($p$) of the available training data. `ExGRPO` consistently demonstrates superior sample efficiency. Across all datasets and at every data fraction, our method achieves higher accuracy than SFT. Notably, for instance, on SQA and CSQA, `ExGRPO` with just **10%** of the data already outperforms SFT with 100% of the data. This trend is also evident in datasets. This signifies that our approach not only achieves higher peak performance but also learns more effectively from limited data augmented through EI, making it a more efficient distillation strategy.

**Token Efficiency.** Figure 5 compares the average test-time accuracy against the average token count per sample used during training for `ExGRPO` and baseline methods. `ExGRPO` generates a comparable number of tokens across baselines during inference. Notably, `ExGRPO` significantly outperforms the general trendline established by the baseline methods, indicating a more effective use of training tokens for performance gain, suggesting that the richer interactions facilitated by EI and our RL framework lead to more impactful learning per token.

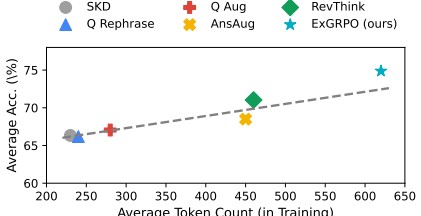

Figure 5: Average accuracy *v.s.* average training token count. The dashed line shows the regression over the baselines.

## 4.7 CASE STUDY

Figure 6 illustrates an example from the MATH dataset requiring continuity at $x = -2$ and $x = 2$. While the ground truth solution yields $a = -3$, $b = 3$, and $a + b = 0$, the base model (Gemma-7b-it) fails by repeating template-like "limit" statements without enforcing the necessary equalities. This reflects *pattern following* rather than true constraint reasoning. By contrast, EI probes reformulate the task into explanation-seeking questions (e.g., "Which two pieces must be equal at $x = 2$?"). These encourage explicit matching of adjacent pieces, guiding both the teacher and the student to recover the correct logic and solution. Moreover, with ExGRPO, rewards are tied not only to final correctness but also to maintaining a coherent reasoning trajectory across probes. This enforces alignment to structural steps, helping the student generalize beyond rote imitation.

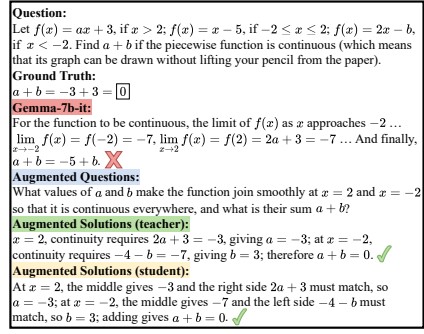

**Question:**
Let $f(x) = ax + 3$, if $x > 2$; $f(x) = x - 5$, if $-2 \le x \le 2$; $f(x) = 2x - b$, if $x < -2$. Find $a + b$ if the piecewise function is continuous (which means that its graph can be drawn without lifting your pencil from the paper).
**Ground Truth:**
$a + b = -3 + 3 = \boxed{0}$
**Gemma-7b-it:**
For the function to be continuous, the limit of $f(x)$ as $x$ approaches $-2$ …
$\lim_{x \to -2} f(x) = f(-2) = -7$, $\lim_{x \to 2} f(x) = f(2) = 2a + 3 = -7$ … And finally,
$a + b = -5 + b$. ✗
**Augmented Questions:**
What values of $a$ and $b$ make the function join smoothly at $x = 2$ and $x = -2$ so that it is continuous everywhere, and what is their sum $a + b$?
**Augmented Solutions (teacher):**
$x = 2$, continuity requires $2a + 3 = -3$, giving $a = -3$; at $x = -2$, continuity requires $-4 - b = -7$, giving $b = 3$; therefore $a + b = 0$. ✓
**Augmented Solutions (student):**
At $x = 2$, the middle gives $-3$ and the right side $2a + 3$ must match, so $a = -3$; at $x = -2$, the middle gives $-7$ and the left side $-4 - b$ must match, so $b = 3$; adding gives $a + b = 0$. ✓

Figure 6: An example in dataset MATH.

Figure 7: Distillation performance using `ExGRPO` with Llama-3-70B-Instruct and Gemini-1.5-Pro teachers.

| Model Setting | Teacher | SQA | CSQA | ARC-c | GSM8K | Avg |
|---|---|---|---|---|---|---|
| | *Teacher Models* | | | | | |
| Teacher (Zero-shot) | Llama-3-70B | 76.8 | 80.8 | 90.5 | 92.0 | 85.0 |
| Teacher (Zero-shot) | Gemini-1.5-Pro | 78.6 | 77.9 | 92.0 | 94.5 | 85.8 |
| | *Qwen2.5-7B (Student Model)* | | | | | |
| Zero-shot (Baseline) | N/A | 72.9 | 71.3 | 85.4 | 90.9 | 77.9 |
| ExGRPO (Ours) | Llama-3-70B | 77.2 | 79.4 | 90.1 | 91.2 | 80.9 |
| ExGRPO (Ours) | Gemini-1.5-Pro | **79.0** | **80.9** | **91.5** | 91.5 | **82.5** |
| | *Gemma-7B (Student Model)* | | | | | |
| Zero-shot (Baseline) | N/A | 56.3 | 66.3 | 68.3 | 41.1 | 46.8 |
| ExGRPO (Ours) | Llama-3-70B | 67.5 | 75.1 | 78.2 | 63.4 | 65.3 |
| ExGRPO (Ours) | Gemini-1.5-Pro | **69.4** | **76.8** | **79.9** | **65.3** | **67.2** |

## 4.8 COMPATIBILITY WITH OPEN-SOURCE TEACHER MODEL

A common question raised was the reliance on the proprietary Gemini-1.5-Pro teacher. To address this concern, we conducted an additional set of experiments using the fully open-source Meta-Llama-3-70B-Instruct model as the teacher. As shown in Table 7, ExGRPO remains highly effective even under this alternative setup. When distilling into Gemma-7B, ExGRPO with the Llama-3 teacher yields an average improvement of +18.5% over the zero-shot baseline, demonstrating that the method is robust and continues to provide substantial gains regardless of teacher choice. Although Llama-3 is slightly weaker than Gemini-1.5 on some challenging reasoning tasks, it performs competitively on ARC-c and GSM8K, resulting in distilled student performance that is very close to the Gemini-based results (typically within 1–2%). This confirms that the gains arise primarily from the ExGRPO training algorithm rather than from any specific advantage of the proprietary teacher model. Overall, these findings show that ExGRPO is reproducible, scalable, and practical even when using widely accessible open-weight models.

## 5 CONCLUSION

We introduce `ExGRPO`, a novel distillation framework that enhances student LLM reasoning by integrating Explanatory Inversion (EI) with a tailored rule-based reinforcement learning algorithm. By using EI to generate diverse "explanatory probes" and leveraging a Dialogue Structure Utility Bonus, `ExGRPO` fosters deeper engagement with the reasoning process beyond simple imitation. Our experiments confirm substantial improvements in student model performance, sample efficiency, and OOD generalization, marking a promising step towards more capable and reliable distilled LLMs.

## ACKNOWLEDGEMENT

This research was funded by the National Science Foundation (NSF) under grant III-2229461. The views and conclusions contained in this document are those of the authors and should not be interpreted as representing official policies, either expressed or implied, of NSF.

## ETHICS STATEMENT

The primary contribution of this work is the development of methods to create more computationally efficient and robust reasoning language models. The positive societal impact of this research includes enhancing the accessibility of advanced AI capabilities. By enabling smaller models to perform complex reasoning, our work can lower the computational and financial barriers to entry, allowing for wider use in beneficial applications such as education and scientific research, particularly in resource-constrained environments. Furthermore, the use of more efficient models can contribute to reducing the overall energy consumption associated with AI technologies.

However, we also recognize potential risks. A core ethical consideration is the propagation of biases from the teacher model to the student. As with all distillation methods, any societal biases, factual inaccuracies, or harmful stereotypes present in the teacher model can be inherited and potentially amplified by the student. Additionally, the creation of more powerful and accessible reasoning models raises concerns about potential misuse, such as the large-scale generation of sophisticated misinformation or deceptive content.

To mitigate these risks, we advocate for continued research into bias detection and mitigation techniques specifically tailored for distilled models. We also recommend that practitioners deploying models trained with our method conduct thorough evaluations for fairness and safety in their specific use cases. We believe that transparently acknowledging these limitations and encouraging responsible development practices are crucial steps toward ensuring the positive impact of this line of research.

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

CONTENTS OF THE APPENDIX

## A   COMPREHENSIVE REVIEW OF RELATED WORK

**Knowledge Distillation for LLMs.** Knowledge Distillation (KD) has become a pivotal technique for compressing large language models (LLMs) into more compact and computationally efficient counterparts (Hinton et al., 2015; Xu et al., 2024b; Zhang et al., 2025b). Most of the existing KD methods focus on off-policy settings. This means the teacher's reasoning generation is usually decoupled from the student training. Initial KD methodologies predominantly centered on transferring knowledge through the alignment of the teacher model's logits or hidden states (Sanh et al., 2019; Jiao et al., 2019). With the ascent of prompting (Wei et al., 2022; White et al., 2023; Beigi et al., 2024) and instruction-tuning paradigms (Zhang et al., 2023; Zhou et al.), subsequent research has adapted KD to imbue student models with reasoning rationales and instruction-following capabilities, often by directly learning from the outputs of teacher LLMs (Fu et al., 2023; Li et al., 2023; West et al., 2022; Chiang et al., 2023; Hsieh et al., 2023; Tan et al., 2024; 2025; Li et al., 2025b; Tian et al., 2025; Lei et al., 2025; Huang et al., 2026). For instance, the Distilling step-by-step methodology (Hsieh et al., 2023) extracts CoT rationales from the teacher model, which are then employed as supplementary supervision within a multi-task training framework. Similarly, Orca (Mukherjee et al., 2023), WizardLM (Xu et al., 2024a), and MiniChat (Li et al., 2025d) utilize synthetic datasets generated by teacher models via prompting and use next token prediction as the training objective. CoT-Evo (Feng et al., 2025) utilizes the evolutionary strategy to enhance the distillation. Agarwal et al. (2024) further introduces an on-policy distillation approach that enhances distillation by leveraging the on-the-fly discrepancies between the student's and teacher's reasoning processes. Extending this line of research, we propose to leverage EI probes to elicit and distill reasoning from teacher models.

**Challenges of Distilled LLMs.** Distilled LLMs face challenges limiting their practical utility. Commonly cited limitations include exposure bias due to complying with teacher-derived supervision targets (Xu et al., 2019; Chu et al., 2025), inability to capture multi-teacher learning (Tian et al., 2025; Chen et al., 2024) or generate long-context reasoning (Yan et al., 2025; Chen et al., 2025b; Yeo et al., 2025) caused by teacher-student discrepancies, and inefficiencies in internalizing complex preference signals from the teacher (Gu et al., 2025; Zhang et al., 2025d; Gu et al., 2025). Additionally, recent studies have noted that distilled models exhibit limitations specific to reasoning-intensive tasks, including shallow reasoning depth (Chen et al., 2025d; Yang et al., 2025), inconsistent multi-step reasoning (Chen et al., 2025c; Guo et al., 2025b), and logical coherence issues (Li et al., 2025a;c; Magister et al., 2022; Zelikman et al., 2022). The above-mentioned limitations undermine the generalization ability of distilled LLMs. Seminal work such as RevThink (Chen et al., 2025a) has shown that integrating forward and backward reasoning can improve performance on downstream tasks. Nevertheless, a fundamental question remains: how to advance the generalization capabilities of distilled student models, beyond rote memorization and towards contextual understanding.

**Rule-based RL for LLMs.** Reinforcement learning (RL) has recently been explored as a promising approach to refine LLMs' generalizability (Schulman et al., 2017; Rafailov et al., 2023; Chu et al., 2025; Shao et al., 2024). Recent advancements in rule-based RL approaches, such as GRPO, have demonstrated that even coarse, outcome-only rewards can elicit strong reasoning behavior (Guo et al., 2025a; Xie et al., 2025). Moreover, multi-turn RL methods exploit dialogue interactions to reinforce intermediate reasoning coherence (Zhou et al., 2024; Shani et al., 2024; Zhang et al., 2025a). Inspired by these advancements, our proposed Explanatory GRPO (`ExGRPO`) algorithm uniquely extends rule-based RL for distillation. A key challenge is that standard GRPO is ***ill-suited*** for this task, as its outcome-only reward cannot supervise the intermediate reasoning steps present in teacher-generated traces. We address this limitation by introducing the novel Dialogue Structure Utility (DSU) reward, which is co-designed with our Explanatory Inversion (EI) framework. This synergy allows `ExGRPO` to explicitly reward coherent reasoning across the multi-turn explanatory dialogues generated by EI, compelling the student to internalize the reasoning process rather than merely imitating final outputs.

## B   DISCUSSION ON POTENTIAL LIMITATIONS

While our proposed `ExGRPO` framework demonstrates significant improvements in reasoning distillation, we identify several limitations that offer avenues for future research and are important for contextualizing our contributions.

- **Dependence on Teacher Model Quality:** The efficacy of our Explanatory Inversion (EI) probe generation is inherently tied to the capabilities of the teacher model (Gemini-1.5-Pro in our

experiments). While the structured nature of our probe templates provides benefits even with weaker teachers, the quality and nuance of the generated explanatory dialogues are undoubtedly influenced by the teacher's reasoning abilities. Errors, biases, or logical gaps in the teacher's outputs could be propagated to the student model. Future work will include experiments with a wider range of open-source and less capable teacher models to quantify this dependency.

- **Manual Design of Explanatory Probes:** The ten categories of EI probes were inspired by principles from cognitive science but were ultimately manually designed. While we use templates to generalize these probes across different tasks, this approach may not scale perfectly to all possible reasoning domains. We are exploring more automated methods for probe generation, such as learned probe controllers or self-play mechanisms, which could enhance the scalability and adaptability of our framework without altering its core logic.

- **Limited Scope of Out-of-Distribution (OOD) Evaluation:** Our current OOD evaluation primarily focuses on robustness to structural and format-level perturbations (e.g., GSM8K-Reversal) rather than broad semantic or domain-level shifts. This was a deliberate choice to test the specific generalization failures we identified, but we acknowledge that testing on more diverse domains would provide a more comprehensive assessment of the model's generalization capabilities.

## C   FURTHER DETAILS ON METHODOLOGY AND CONTRIBUTIONS

This section provides additional details on key aspects of our methodology to ensure clarity.

- **Contribution Regarding the "Reversal Curse":** Our work's contribution is to systematically identify and address the *amplified generalization limitation* in *distilled* LLMs, for which the Reversal Curse is a primary example. Our research investigates why distillation pipelines can exacerbate such failures (e.g., by encouraging superficial pattern matching) and proposes a solution that moves beyond simple A-to-Q data augmentation toward deeper reasoning alignment.

- **Contribution regarding Reinforcement Learning or GRPO:** A key distinction of our work is the specific design of our RL component. Standard GRPO is incompatible with distillation settings involving teacher reasoning traces. GRPO optimizes the policy using only final answer correctness as the reward signal. It does not incorporate any intermediate reasoning steps or teacher-generated traces, making it unable to supervise the student's reasoning behavior. Our framework introduces a novel Dialogue Structure Utility (DSU) reward, which is crucial for enabling GRPO to reward reasoning consistency across the multi-turn dialogues generated by EI. This allows the student to internalize reasoning behaviors, not just mimic outputs, a fact supported by our ablation studies. Furthermore, applying a DSU-style reward to standard SFT baselines like RevThink would be non-trivial, as they do not produce the coherent, multi-step dialogue structures necessary for such a reward. Our EI probe design and DSU reward function are co-designed to uniquely enable this form of structured reasoning supervision.

- **Computation of the Dialogue Structure Utility Bonus ($r_{dsu}$):** The Dialogue Structure Utility Bonus ($r_{dsu}$) is a purely outcome-based reward and is not based on semantic "agreement" or string matching between different probe answers. A bonus is awarded if the student model's final-answer accuracy for the original question $Q$ is higher after engaging with a full $k$-turn explanatory dialogue compared to a partial $k'$-turn dialogue ($k' < k$). This mechanism incentivizes the model to effectively utilize the entire reasoning scaffold provided by the probes, thereby promoting a deeper internalization of the reasoning process.

- **Use of Explanatory Probes at Inference Time:** The multi-turn explanatory dialogues are a *training-time-only* mechanism. They function as a scaffold to build robust reasoning abilities within the student model. At inference time, the distilled model processes the input question in a single pass without any probes. This design ensures that our method does not introduce any inference-time latency or complexity. The resulting strong single-pass performance indicates that the reasoning structures have been successfully internalized.

- **Order of Probing Questions:** In our training protocol, the order of the $k$ probes used in each dialogue is randomized. This is a deliberate design choice to encourage order-invariant robustness and prevent the model from overfitting to a fixed sequence of explanatory challenges.

- **Clarification on Framework Complexity and Computational Cost:** Most distillation methods, including our baselines, utilize a teacher LLM to augment data. Similarly, data filtering and

supervised warm-up are standard in RL-based LLM pipelines. The primary additional step in our framework is the `ExGRPO` reinforcement learning stage. Therefore, the overall complexity and training cost are comparable to existing practices (Guo et al., 2025a). Our design is modular: EI generation is a one-time, offline process; filtering and warm-up are lightweight; and RL training is efficient on curated datasets (1–3k examples). In practice, `ExGRPO` is approximately 1.5x more compute-intensive than a single-stage SFT baseline while achieving significantly better generalization. Critically, inference remains as efficient as the baselines, as the multi-turn probing is a training-only scaffold.

## D  IMPLEMENTATION DETAIL

**Models.** The teacher model used for generating Explanatory Inversion (EI) probes is Gemini-1.5-Pro. Student models used in our main experiments are Gemma-7b-it and Qwen2.5-7b-instruct, initialized from their official pre-trained weights from Huggingface. The reference model ($\pi_{\text{ref}}$) used for KL regularization in the `ExGRPO` objective (Equation (7) in the main paper) is the student model obtained after Stage 1 SFT (Section 3.3 in the main paper).

**Training Procedure.** *Stage 2 (SFT):* Student models are fine-tuned on the curated $D_{EI}$ dataset. We use the AdamW optimizer with a learning rate of $2 \times 10^{-5}$, a batch size of 32, and train for $P$ epochs (e.g., $P = 1$ or $P = 3$ as per ablation studies).

*Stage 3 (`ExGRPO`):* The SFT-warmed-up student model is further trained using our `ExGRPO` algorithm. We use the AdamW optimizer with a learning rate of $1 \times 10^{-6}$ and a batch size of 16. The RL phase runs for 1 epoch. For `ExGRPO`, the group size $G$ for generating trajectories (for both Scenario A and Scenario B) is set to 4. The clipping hyperparameter $\epsilon$ (Equation (7) in the main paper) is set to 0.2. The KL regularization coefficient $\beta$ is set to 0.01. For the Dialogue Structure Utility Bonus ($r_{\text{dsu}}$), the number of EI turns $k$ in the Full Dialogue (Scenario A) is set to 5. For the Partial Dialogue (Scenario B), $k'$ is set to 2. The bonus value $\delta$ is 0.1 and the performance margin $\nu$ is 1.05. The imitation-based policy regularization loss $\mathcal{L}_{\text{SFT-aux}}$ (Equation (8) in the main paper) is applied with a weight of [Specify SFT-aux weight, e.g., 0.1] relative to the `ExGRPO` loss. Note that we manually add a pair of special tokens **<think>** and **</think>** to wrap the dialog and use a format reward as standard GRPO to strengthen the output of the tokens. We observe that this reward is easy to learn and converge, and thus omit the details in the main contents.

## E  FULL LIST OF EXPLANATORY INVERSION RULES

This appendix details the full set of $N = 10$ Explanatory Inversion rule categories used in our work, complementing the subset presented in Section 3.1. Illustrative examples for each category are provided below.

**R1. Why-Based Transformation ($f_1$):** Transforms a descriptive aspect of $Q$ or a step $s_j \in R_T$ into an inquiry about its underlying reasons or justifications. Formally, if $Q \to A$ or $s_j \vdash s_{j+1}$, then $Q_1^{\text{aug}} \approx \text{Why}(Q \to A \text{ via } R_T)$ or $\text{Why}(s_j \vdash s_{j+1})$. This aligns with cognitive drives for explanation-seeking and understanding causality (Keil, 2006).

**R2. Causal Relationship Probing ($f_2$):** Rephrases elements of $Q$ or $R_T$ to explicitly elicit or verify directional cause-effect chains. If $R_T$ implies $C \Rightarrow E$, then $Q_2^{\text{aug}} \approx \text{HowDoes}(C \text{ lead to } E)?$. This taps into fundamental causal cognition (Sloman & Sloman, 2009).

**R3. Process and Mechanism Elucidation ($f_3$):** Generates probes requiring detailed, step-by-step explanations of processes or mechanisms implied or stated in $R_T$. $Q_3^{\text{aug}} \approx \text{DescribeProcess}(P \text{ in } R_T)$. This relates to forming and querying mental models of systems (Machamer et al., 2000).

**R4. Counterfactual Scenario Generation ($f_4$):** Creates probes exploring outcomes if a premise $p$ within $Q$ or $R_T$ were altered ($\neg p$). $Q_4^{\text{aug}} \approx \text{WhatIf}(\neg p, Q, R_T)?$. This is central to counterfactual thinking and robust understanding (Byrne, 2007).

**R5. Comparative and Contrastive Framing ($f_5$):** Reformulates questions to require comparison or contrast between concepts $C_1, C_2$ (from $Q, A, R_T$) or conditions. $Q_5^{\text{aug}} \approx$

Compare($C_1, C_2$, wrt dimension $D$). This engages analogical and comparative reasoning faculties (Gentner, 1983).

**R6. Hypothesis Generation and Evaluation ($f_6$):** Generates questions prompting the model to propose or evaluate alternative hypotheses or explanations $H'$ for $A$ given $Q$. $Q_6^{\text{aug}} \approx$ EvaluateAlternative($H', Q \rightarrow A$). This mirrors scientific inquiry and hypothesis testing in cognition (Klahr & Dunbar, 1988).

**R7. Applied Scenario Generalization ($f_7$):** Transforms $Q$ or concepts in $R_T$ into new, related practical scenarios to test generalization. If $R_T$ uses concept $C$, $Q_7^{\text{aug}} \approx$ Apply($C$, NewScenario $S'$)?. This connects to situated cognition and transfer of learning (Lave & Wenger, 1991).

**R8. Multi-Step Reasoning Decomposition/Construction ($f_8$):** Probes understanding of complex reasoning chains by asking for intermediate steps, or the logical antecedents/consequents of a step $s_j \in R_T$. $Q_8^{\text{aug}} \approx$ WhatPrecedes($s_j$)? or WhatFollows($s_j$)?. This relates to problem decomposition and planning (Newell et al., 1972).

**R9. Temporal or Sequential Dynamics Exploration ($f_9$):** If $R_T$ involves a sequence or temporal evolution, probes are generated to question the order, dependencies, or changes over time/steps. $Q_9^{\text{aug}} \approx$ HowEvolves($X$, time/sequence in $R_T$)?. This taps into temporal reasoning and narrative comprehension (Zacks et al., 2007).

**R10. Direct Explanatory Challenge of Reasoning Steps ($f_{10}$):** Poses questions directly asking for justification of specific assertions or logical transitions $s_j \rightarrow s_k$ within $R_T$. $Q_{10}^{\text{aug}} \approx$ Justify($s_j \rightarrow s_k$ in $R_T$). This encourages metacognitive reflection and articulable understanding (Chi et al., 1989).

# F    ANALYSIS OF INDIVIDUAL EXPLANATORY INVERSION (EI) RULE CONTRIBUTIONS

To understand the relative importance of different probe types, we performed a post-hoc analysis where the Qwen2.5-7B-Instruct student model was fine-tuned using augmented data from each of our 10 EI rule categories in isolation. We compare these against a standard "Rephrase Question" augmentation baseline. Table 4 presents the accuracy of the resulting models on four diverse reasoning datasets.

The results show that most EI rule, when applied individually, outperforms the rephrasing baseline, underscoring the general effectiveness of our cognitively-inspired approach. However, certain probe types are clearly more impactful. Notably, **Counterfactual (R1)**, **Why-Based (R10)**, and **Decomposition (R2)** probes yield the highest average performance gains. This suggests that compelling a model to reason about alternative realities, question the fundamental basis of statements, and break down complex problems are particularly effective strategies for enhancing its reasoning capabilities.

The data also reveals task-specific affinities. For instance, Decomposition (R2) shows exceptional strength on the logic-heavy GSM8K dataset, while Counterfactual (R1) excels on the commonsense-oriented SQA and ARC-c tasks. No single rule dominates across all benchmarks, reinforcing our main finding that the combination of all 10 EI rules provides the most robust and significant performance improvement, achieving the highest scores across all datasets.

# G    DATASETS

In this subsection, we provide detailed descriptions of the datasets used in our experiments, encompassing both the primary training datasets and the out-of-distribution (OOD) evaluation datasets. Dataset statistics, including licensing, number of training samples (original and filtered), and test sizes, are provided in Table 5.

## G.1    TRAINING DATASETS

**Commonsense Reasoning:**

Table 4: Performance (Accuracy %) of SFT with single EI probe types on Qwen2.5-7B-Instruct. All EI rules outperform the baseline, and the full EI combination yields the best results.

| Probe Type (Rule) | SQA | ARC-c | GSM8K | ANLI | Avg. |
|---|---|---|---|---|---|
| Rephrase Question (Yu et al., 2024) | 75.3 | 89.1 | 87.2 | 63.9 | 78.88 |
| Counterfactual (R1) | 76.8 | 89.5 | 91.2 | 63.1 | 80.15 |
| Decomposition (R2) | 76.1 | 88.9 | 90.5 | 63.5 | 79.75 |
| Justification (R3) | 75.7 | 88.3 | 89.8 | 62.8 | 79.15 |
| Analogy-Based (R4) | 74.9 | 87.1 | 88.4 | 61.7 | 78.03 |
| Contrastive (R5) | 75.3 | 87.5 | 88.6 | 62.1 | 78.38 |
| Hypothesis Gen. (R6) | 75.5 | 88.1 | 89.5 | 62.5 | 78.90 |
| Causal Probing (R7) | 76.0 | 88.5 | 90.1 | 63.0 | 79.40 |
| Generalization (R8) | 74.5 | 86.9 | 88.0 | 61.5 | 77.72 |
| Temporal Dynamics (R9) | 74.8 | 87.0 | 88.2 | 61.9 | 77.98 |
| Why-Based (R10) | 76.5 | 89.2 | 90.8 | 63.3 | 79.95 |
| **Full EI (All 10 rules)** | **79.0** | **91.5** | **91.5** | **64.0** | **81.50** |

- **StrategyQA (SQA)** (Geva et al., 2021): A dataset of yes/no questions that require implicit multi-hop reasoning. It contains 2,061 training samples, of which 1,544 are retained after EI filtering, and 229 test samples. Licensed under MIT.

- **CommonsenseQA (CSQA)** (Talmor et al., 2019): This dataset includes 9,741 original multiple-choice questions with rich commonsense demands; 6,478 filtered training examples and 1,140 test examples are used in our experiments. Licensed under MIT.

- **ARC-Challenge (ARC-c)** (Clark et al., 2018): A benchmark of 1,199 challenging grade-school science questions. After filtering, 1,035 are retained for training, with 1,172 samples used for testing. Licensed under CC BY-SA 4.0.

**Mathematical Reasoning:**

- **GSM8K** (Cobbe et al., 2021): Comprises 7,379 grade-school math problems written by professional educators. 4,293 filtered training samples and 1,339 test questions are used. Licensed under MIT.

- **MATH** (Hendrycks et al., 2021): A competition-level dataset with 7,500 original math problems, 2,511 filtered training samples, and 5,000 test samples. Licensed under MIT.

**Tabular Data Reasoning:**

- **TabMWP** (Lu et al., 2022): Contains 23,059 math word problems grounded in tables. After filtering, 15,544 training samples remain, with 7,686 samples used for testing. Licensed under CC BY-SA 4.0.

**Natural Language Inference:**

- **ANLI (Round 3)** (Nie et al., 2020): A large-scale NLI dataset collected via an adversarial human-in-the-loop process. We randomly sample 2,000 examples from the original 100,459, of which 883 remain after filtering. We use 1,200 samples for testing. Licensed under CC BY-NC 4.0.

**Logical Reasoning:**

- **Date Understanding** (Srivastava et al., 2022): This dataset challenges models with temporal reasoning tasks about specific dates. We randomly split the small dataset into 200 training samples (all retained) and 169 test samples. Licensed under Apache.

## G.2  OUT-OF-DISTRIBUTION (OOD) EVALUATION DATASETS

To evaluate the generalization capabilities of our models, we employ the following OOD datasets. These are held out during training and used exclusively for testing:

- **BoolQ** (Clark et al., 2019): A dataset of 3,270 naturally occurring yes/no questions, designed to reflect real-world information-seeking queries. No training samples are used. Licensed under CC BY-SA 3.0.
- **OpenBookQA** (Mihaylov et al., 2018): Features 500 science questions that test both factual recall and application. No training samples are used. Licensed under Apache.
- **e-SNLI** (Camburu et al., 2018): An extended version of SNLI with 9,824 test samples, augmented with natural language explanations. No training data is included in our setting. Licensed under CC BY-NC 4.0.
- **GSM8K-Reversal** (Guo et al., 2024): A reversed-format variant of GSM8K, assessing the model's ability to infer inputs from outputs. We evaluate using 777 test samples without training. Licensed under Apache.

Table 5: The datasets used in this work are listed in the order of appearance. For each dataset, we report the domain, the number of original training samples, the number of filtered training samples, and the number of testing samples. Note that the last four datasets are held out and thus contain no filtered training samples. Due to the large size of ANLI's training set, we randomly sampled 2,000 instances, of which 883 remained after filtering. For the Date Understanding dataset, given its small size, we randomly split the data into 200 training and 169 testing samples, and we keep all the training data.

| Dataset | Domain | License | Train (Original) | Train (Filtered) | Test |
|---|---|---|---|---|---|
| SQA (Geva et al., 2021) | Commonsense | MIT | 2061 | 1656 | 209 |
| CSQA (Talmor et al., 2019) | Commonsense | MIT | 9741 | 7770 | 1221 |
| ARC (Clark et al., 2018) | Commonsense | CC BY-SA 4.0 | 1199 | 1080 | 1172 |
| MATH (Hendrycks et al., 2021) | Math | MIT | 7500 | 6228 | 5000 |
| GSM8K (Cobbe et al., 2021) | Math | MIT | 7379 | 7085 | 1319 |
| TabMWP (Lu et al., 2022) | Math (Tabular) | CC BY-SA 4.0 | 23 059 | 22 802 | 7684 |
| ANLI (r3) (Nie et al., 2020) | NLI | CC BY-NC 4.0 | 100 459 | 1584 | 1200 |
| Date (Srivastava et al., 2022) | Logic | Apache | 200 | 185 | 169 |
| BoolQ (Clark et al., 2019) | Commonsense | CC BY-SA 3.0 | 9427 | 0 | 3270 |
| OpenbookQA (Mihaylov et al., 2018) | Commonsense | Apache | 4957 | 0 | 500 |
| e-SNLI (Camburu et al., 2018) | NLI | CC BY-NC 4.0 | 549 367 | 0 | 9824 |
| GSM8K-Rev (Guo et al., 2024) | Math | Apache | – | 0 | 777 |

## H  BASELINE METHODS

We compare `ExGRPO` with a diverse suite of baseline methods spanning three main categories:

**(1) Zero-shot Reasoning:** As a reference point, we evaluate the *zero-shot* performance of the student models by prompting them to directly answer questions without any fine-tuning. This follows the CoT prompting paradigm introduced in (Kojima et al., 2022), which reveals the model's native reasoning capability when provided with a task-specific instruction and an example prompt.

**(2) Knowledge Distillation:** We include baselines that perform knowledge distillation from a stronger teacher model to a student model using chain-of-thought (CoT) reasoning traces:

- **Symbolic Knowledge Distillation (SKD)** (Li et al., 2023; West et al., 2021): This method extracts symbolic reasoning traces (i.e., CoT rationales) from the teacher model and trains the student using a next-token prediction objective on the rationale, followed by supervision on the final answer.
- **Distilling Step-by-Step** (Hsieh et al., 2023): In addition to standard next-token prediction on the CoT rationale, this method introduces an auxiliary loss term to supervise the student's final answer directly. The goal is to jointly distill both intermediate reasoning steps and the conclusive decision.

**(3) Data Augmentation:** This category of baselines augments the training set with additional samples generated by prompting a teacher model, while maintaining the standard next-token prediction loss on the rationale and answer:

- **Question Rephrasing** (Yu et al., 2023): The teacher model is prompted to paraphrase an original question, yielding a semantically equivalent variant with the same answer and reasoning rationale. This increases input diversity and reduces overfitting.
- **Question Augmentation** (Li et al., 2024): The teacher is asked to generate new, logically related questions based on the original input, allowing the model to generalize across similar yet distinct reasoning scenarios.
- **Answer Augmentation** (Yu et al., 2023): For each original question, the teacher generates alternative correct reasoning paths (CoTs) that arrive at the same answer. This provides multi-rationale supervision and improves robustness to reasoning variation.
- **RevThink** (Chen et al., 2025a): A recent augmentation strategy where the teacher model is queried with reversed formulations of the original questions. These backward-style prompts yield contrastive reasoning samples that encourage deeper conceptual alignment during student training.

All baseline methods are implemented using the same student architectures (Gemma-7B-it and Qwen2.5-7B-Instruct) and trained under consistent settings to ensure fair comparison with our proposed `ExGRPO` method.

# I    HYPERPARAMETERS

For the training procedure, we adopt a three-stage training pipeline:

**Stage 1 (EI Generation):** EI probes are generated from the Gemini teacher model using 10 predefined categories (Appendix E).

**Stage 2 (Supervised Fine-Tuning):** The student models are fine-tuned on the curated EI-augmented dataset $D_{EI}$.

- Optimizer: AdamW
- Learning Rate: $2 \times 10^{-5}$
- Batch Size: 32
- Epochs: $P = 1$ or $P = 3$ (depending on ablation)

**Stage 3 (`ExGRPO`):** The student is further optimized via our `ExGRPO` algorithm using reward-based training.

- Optimizer: AdamW
- Learning Rate: $1 \times 10^{-6}$
- Batch Size: 16
- Epochs: 1
- Group Size $G$: 4
- KL Regularization Coefficient $\beta$: 0.01
- Clipping Coefficient $\epsilon$: 0.2

**Dialogue Structure Utility Bonus (DSU).**    This reward-shaping mechanism is used to encourage structured explanatory probes. We perform parameter search on variables:

- EI Turns in Full Dialogue ($k$): 5
- EI Turns in Partial Dialogue ($k'$): 2
- Bonus Value ($\delta$): {0.1, **0.2**, 0.5, 1.0}
- Performance Margin ($\nu$): {0.50, 1.00, **1.05**, 1.10, 1.15}

**Infrastructure.** All experiments are run on 8 NVIDIA A100 GPUs with 80GB RAM using the Huggingface Transformers library.

## J  PROOF OF THEOREM 3.1

**Formal Assumptions.** We state the assumptions required for the theoretical guarantee:

**A1** (*Finite-horizon POMDP*) The explanatory probe process is modeled as a finite-horizon POMDP $M = (O, C, A, T, \mu_1, R, N)$, where $O$ are observations, $C$ hidden context, $A$ actions, $T$ transitions, $\mu_1$ initial state distribution, $R$ the reward function, and $N$ the horizon.

**A2** (*Support overlap*) $\pi_{\text{new}}$ and $\pi_k$ have overlapping support, so importance ratios $\rho^{(g)}$ are well defined.

**Theorem 3.1.** *Let $\pi_k$ and $\pi_{k'}$ be student policies trained with full ($k$-turn) and partial ($k' < k$) explanatory probe sequences, respectively. Suppose the Dialogue Structure Utility Bonus $r_{\text{dsu}}$ is applied only when*

$$\mathbb{E}_{\pi_k}[R_{\text{outcome}}(Q, A)] > \nu \cdot \mathbb{E}_{\pi_{k'}}[R_{\text{outcome}}(Q, A)], \quad \text{for some } \nu \geq 1. \tag{11}$$

*Then, under the ExGRPO update with clipped importance sampling and KL regularization, the updated policy $\pi_{\text{new}}$ satisfies*

$$\mathbb{E}_{\pi_{\text{new}}}[R_{\text{outcome}}(Q, A)] \geq \mathbb{E}_{\pi_k}[R_{\text{outcome}}(Q, A)], \tag{12}$$

*with strict inequality whenever $r_{\text{dsu}} > 0$ applies to a nonzero-probability set of trajectories.*

*Proof.* We first show that reward shaping induces a positive advantage shift, then appeal to conservative policy improvement to obtain a monotonic increase in shaped return, and finally translate this into outcome return.

**Step 1. POMDP formulation.** By Assumption A1, the explanatory probe process is modeled as a finite-horizon partially observable Markov decision process (POMDP):

$$M = (O, C, A, T, \mu_1, R, N),$$

where $O$ denotes the observable history (probes and responses), $C$ denotes hidden context such as ground-truth answers, $A$ denotes student actions, $T$ the transition, $\mu_1$ the initial distribution, $R$ the reward function, and $N$ the horizon. At each turn $t$, the agent observes $o_t$, produces an action $a_t$, and transitions to $o_{t+1}$.

**Step 2. Value, Q-, and advantage functions.** For a policy $\pi$:

$$Q^\pi(o_t, a_t, c) = \mathbb{E}_\pi\Big[ \sum_{t'=t}^{N} r(o_{t'}, a_{t'}, c) \Big], \tag{13}$$

$$V^\pi(o_t, c) = \mathbb{E}_{a_t \sim \pi}[Q^\pi(o_t, a_t, c)], \tag{14}$$

$$A^\pi(o_t, a_t, c) = Q^\pi(o_t, a_t, c) - V^\pi(o_t, c). \tag{15}$$

We focus on the *outcome reward*:

$$R_{\text{outcome}}(Q, A) = \mathbf{1}\{\text{student's final answer matches } A\}. \tag{16}$$

**Step 3. Reward shaping with $r_{\text{dsu}}$.** Define the total shaped reward as

$$R_{\text{total}} = R_{\text{outcome}} + r_{\text{dsu}}, \tag{17}$$

where $r_{\text{dsu}} = \delta > 0$ is applied only when the full $k$-turn sequence outperforms a partial $k'$-turn sequence by a factor $\nu$. By construction,

$$R_{\text{total}} \geq R_{\text{outcome}} \quad \text{pointwise.} \tag{18}$$

Let $J_{\text{out}}(\pi) = \mathbb{E}_\pi[R_{\text{outcome}}]$ and $J_{\text{tot}}(\pi) = \mathbb{E}_\pi[R_{\text{total}}]$. Clearly $J_{\text{tot}}(\pi) \geq J_{\text{out}}(\pi)$.

**Step 4. Decomposition.** Because $r_{\text{dsu}}$ applies only to outcome-correct trajectories in Scenario A, we can write

$$J_{\text{tot}}(\pi) = J_{\text{out}}(\pi) + \delta \cdot p_A(\pi), \tag{19}$$

where $p_A(\pi) = \Pr_\pi[\text{Scenario A}, B, R_{\text{outcome}} = 1]$.

**Step 5. Advantage shift.** On trajectories where $r_{\text{dsu}} > 0$, the shaped advantage satisfies

$$A_{\text{tot}}(o_t, a_t, c) = A_{\text{out}}(o_t, a_t, c) + \delta, \tag{20}$$

so outcome-correct actions are strictly more advantageous than without shaping.

**Step 6. Conservative policy improvement.** ExGRPO optimizes the clipped surrogate objective:

$$\mathcal{L}_{\text{ExGRPO}}(\theta) = \mathbb{E}_{\tau \sim \pi_k} \left[ \sum_{g=1}^{G} \min\left( \rho^{(g)} U^{(g)}, \text{clip}(\rho^{(g)}, 1-\epsilon, 1+\epsilon) U^{(g)} \right) \right] - \beta D_{\text{KL}}(\pi_\theta \,||\, \pi_{\text{ref}}), \tag{21}$$

where $U^{(g)}$ are group-normalized advantages based on $R_{\text{total}}$. Here, Assumption A2 guarantees that the importance ratios $\rho^{(g)}$ are well-defined because $\pi_{\text{new}}$ and $\pi_k$ share overlapping support. Together with the KL penalty, this ensures the surrogate objective satisfies the conditions of conservative policy improvement (as in PPO/TRPO theory (Schulman et al., 2017)), so that

$$J_{\text{tot}}(\pi_{\text{new}}) \geq J_{\text{tot}}(\pi_k), \tag{22}$$

with strict improvement if there exists a nonzero set with $A_{\text{tot}} > 0$.

**Step 7. From shaped to outcome reward.** Using Step 4,

$$J_{\text{out}}(\pi_{\text{new}}) + \delta p_A(\pi_{\text{new}}) \geq J_{\text{out}}(\pi_k) + \delta p_A(\pi_k). \tag{23}$$

By Step 5, $p_A(\pi_{\text{new}}) \geq p_A(\pi_k)$. Therefore,

$$J_{\text{out}}(\pi_{\text{new}}) \geq J_{\text{out}}(\pi_k). \tag{24}$$

If $r_{\text{dsu}} > 0$ applies with positive probability, the inequality is strict.

**Conclusion.** ExGRPO with the dialogue-structure utility bonus ensures monotonic improvement in outcome accuracy, with strict gains when bonus-triggered trajectories exist. This completes the proof. $\qquad\square$

## K  BROADER IMPACT

This work advances the development of generalizable and efficient reasoning in language models, which has both positive implications and potential risks.

- **Positive Impact:** Our method enables smaller models to acquire strong reasoning capabilities through distillation and augmentation. This promotes efficient deployment in educational, healthcare, and resource-constrained environments.

- **Fairness and Accessibility:** By reducing the reliance on large-scale, task-specific data and enabling transfer to unseen domains, our approach improves model usability in low-resource and non-English settings.

- **Potential Risks:** Enhanced reasoning in compact models may be misused for automated misinformation or deceptive content generation if not properly constrained. Inherited biases from the teacher model may also go unnoticed.

- **Mitigation Strategies:** We recommend the integration of transparency tools (e.g., explanation audits, provenance tracking) and the use of diagnostic evaluations to monitor the integrity of generated reasoning.

## L  QUALITATIVE ANALYSIS AND CASE STUDIES

In this section, we present illustrative examples of EI-augmented samples and model-generated rationales, as well as the prompts used for EI augmentation. These case studies demonstrate that `ExGRPO`-trained models produce more structured, faithful, and task-sensitive reasoning compared to standard SFT and RevThink baselines. For instance, in GSM8K and ARC, the model correctly decomposes multi-step logic and avoids common distractors.

**Example: Teacher and Student CoT in Dataset GSM8K**

**question**: Natalia sold clips to 48 of her friends in April, and then she sold half as many clips in May. How many clips did Natalia sell altogether in April and May?

**answer_key**: "72"
**answer_text**: "72"

**augmented_questions**:
- Why would adding half of 48 to 48 give us the total number of clips Natalia sold in April and May?
- What is the total number of clips Natalia sold, given that her May sales were directly influenced by – and specifically, half the amount of – her April sales of 48 clips?
- What are the steps involved in calculating the total number of clips Natalia sold, considering she sold 48 clips in April and half that amount in May?

**augmented_solutions_teacher**:
- Because Natalia sold 48 clips in April, and half as many in May. "Half as many" means half of the April amount. So, the May sales are half of 48. To find the total sales, we need to add the April sales (48) and the May sales (half of 48) together.
- Natalia sold half as many clips in May as she did in April. Since she sold 48 clips in April, she sold 48 / 2 = 24 clips in May. Add April sales and May sales: 48 + 24 = 72 clips. Therefore, Natalia sold a total of 72 clips.
- Calculate clips sold in May: Natalia sold half as many clips in May as in April. Half of 48 is 24. Calculate total clips: 48 + 24 = 72. So, Natalia sold 72 clips in total.

**augmented_solutions_student**:
- If she sold 48 clips in April and only half that in May, that means 48 / 2 = 24 clips in May, right? So to get the total, I guess we just add them: 48 + 24 = 72?
- Okay, so in April it was 48 clips. If May was only half, then it should be 24 clips. Then 48 plus 24 gives us... 72? That seems to make sense.
- Let me think this through—Natalia sold 48 in April, then half of that in May. Half of 48 is 24. So I just need to add them: 48 + 24. That gives 72, so that's probably the total.

---

**Example: Teacher and Student CoT in Dataset ARC**

**question**: George wants to warm his hands quickly by rubbing them. Which skin surface will produce the most heat?

**choices**:
**text**: ["dry palms", "wet palms", "palms covered with oil", "palms covered with lotion"]
**label**: ["A", "B", "C", "D"]

**answer_key**: "A"
**answer_text**: "dry palms"

**augmented_questions**:
- Why would rubbing wet palms, dry palms, palms covered with oil, or palms covered with lotion produce different amounts of heat?
- How does the presence of water, oil, or lotion on George's palms affect the friction generated when rubbing them together?
- What are the mechanisms of heat generation through friction, and how do the different surface materials affect these mechanisms?

**augmented_solutions_teacher**:
- Dry skin has a relatively high coefficient of friction, leading to more resistance and thus more heat generated. Water, oil, and lotion reduce friction and hence generate less heat.
- Friction converts kinetic energy into heat. Lubricants like water and oil reduce friction, resulting

in lower heat generation. Dry palms produce the most heat.
- Lotion and oil act as lubricants that decrease interlocking between skin surfaces. Dry palms generate more heat due to higher resistance and rough texture.

augmented_solutions_student:
- Dry palms make more heat because they rub more. Things like water or lotion make hands slippery, so less heat happens.
- When you rub dry hands, it gets hot. But if hands are wet or have oil, they slide more and don't make as much heat. So dry palms are best.
- Oil and lotion make your hands smooth, so rubbing doesn't make much heat. Dry hands don't slide as easy, so they get warmer when rubbed.

---

**Example: Final-Turn Student CoT in Dataset ARC during RL**

<think>
1. **Core**: - The core is isolated and has a high pressure and temperature. The intense heat from the impacts would not significantly affect the core composition or structure. Therefore, the core is the least likely recipient of cometary material.
2. **Mantle**: - The mantle can absorb some cometary material, but the amount delivered is relatively small compared to the total volume of the mantle. The mantle's stability and high heat capacity make it resistant to significant compositional changes. Therefore, the mantle is less likely to be the main recipient of cometary material.
3. **Atmosphere**: - The atmosphere is highly dynamic and can quickly cycle and lose significant portions of cometary material. While it can retain some volatiles, the atmosphere's high volume and rapid cycling make it less likely to be the primary recipient of cometary material.
4. **Hydrosphere**: - The hydrosphere is a vast and stable reservoir. The impacts can release water and other volatiles into the atmosphere, which can then condense and form precipitation, contributing to the hydrosphere. The hydrosphere's large volume and stability make it the most likely recipient of cometary material.
Given these insights, the hydrosphere is the most plausible part of the Earth system that received significant amounts of matter from the Late Heavy Bombardment.
</think>
The best answer is D.

---

**Example: Final-Turn Student CoT in Dataset GSM8K during RL**

<think>
1. **April Sales**: - Natalia sold 48 clips in April. This is clearly stated in the problem and serves as the base value for the total calculation.
2. **May Sales**: - The question states that she sold half as many clips in May as she did in April. Half of 48 is calculated as $48 \div 2 = 24$. So, Natalia sold 24 clips in May.
3. **Total Sales**: - To determine the total number of clips sold in April and May, we add the two amounts: 48 (April) + 24 (May) = 72. This sum represents the complete sales over both months.
Given these steps, the total number of clips Natalia sold in April and May is 72.
</think>
The best answer is 72.

## M   FAILURE CASE ANALYSIS

While ExGRPO demonstrates robust performance, analyzing its failure modes provides valuable insights into the remaining challenges in reasoning distillation. Below, we present a case study from the CommonsenseQA dataset where the model exhibits a specific type of reasoning error we term *shallow distractor override*.

**Example: Failure Case on CommonsenseQA**

**Question (Q):** What would you use to remove nail polish?
A. Hammer B. Toothpaste C. Nail polish remover D. Paint thinner

---

**Correct Answer:** C. Nail polish remover

**ExGRPO Student Model Output:**
```
Nail polish is a cosmetic that is
typically removed using a solvent.
While paint thinner can remove paint,
it's not designed for use on the
human body.  Toothpaste is for dental
care.  Therefore, the answer is A.
Hammer.
```

---

**Analysis:** In this example, the student model correctly identifies the function of a solvent and appropriately rules out both "Paint thinner" and "Toothpaste" with sound reasoning. However, it fails to eliminate the most absurd distractor ("Hammer") and incorrectly selects it as the final answer. This indicates a failure to apply its isolated reasoning coherently to the final decision-making process, instead getting overridden by a superficial association or a flaw in its final selection logic.

---

**Associated Explanatory Probe (from EI during training):**
"Why is toothpaste not appropriate for removing nail polish?"

**Student's Response to Probe (during training):**
"Because toothpaste is not strong enough to remove nail polish and is used for teeth."

**Further Analysis:** The model's correct response to the targeted probe demonstrates that it can reason correctly about specific components of the problem in isolation. The failure arises when it must synthesize this understanding and weigh it against illogical distractors in the context of the original multiple-choice question. This highlights the challenge of ensuring that probe-level understanding is robustly transferred to the final answer-selection step, a key area for future improvement.

## N  PROMPT FOR EI AUGMENTATION

**EI Question Augmentation Rules – GSM8K**

**Convert Direct Questions into Why-Based Questions**
**Rule**: Reformulate the question to ask about the reasons or mechanisms behind the original query.
**Example**:
Original: Does the premise entail that the day mentioned is important for religious worship?
Augmented: Why would Sunday being mentioned in the premise entail that it's a day for Christian worship?

**Emphasize Cause-Effect Relationships**
**Rule**: Reframe the question to highlight causal relationships rather than surface-level descriptions.
**Example**:
Original: What does the premise state about the plane crash?
Augmented: How does the statement that all passengers and crew survived establish the

entailment relationship with the hypothesis about no children being killed?

### Focus on Processes or Mechanisms
**Rule**: Augment questions to ask about the step-by-step processes involved in a phenomenon.
**Example**:
Original: How does the premise information about Kit Kat in Japan relate to the hypothesis?
Augmented: What is the inferential process by which we can conclude Japanese people like Kit Kat based on the premise about product variety and continued production since 1973?

### Include Counterfactual Scenarios
**Rule**: Reformulate the question to explore what would happen if a certain condition were different.
**Example**:
Original: Is the hypothesis about survival entailed by the premise?
Augmented: How would the entailment relationship change if the premise stated that 'most' rather than 'all' passengers survived the crash?

### Highlight Comparisons and Contrasts
**Rule**: Augment the question to compare different cases or conditions.
**Example**:
Original: What relationship exists between the premise about technology articles and the hypothesis about Sunday?
Augmented: How does the inference that Sunday is a Christian worship day differ from other possible inferences about Sunday mentioned in the technology news premise?

### Encourage Hypothesis Exploration
**Rule**: Reformulate the question to ask about possible explanations or hypotheses.
**Example**:
Original: Why is there an entailment between the Kit Kat production and Japanese preferences?
Augmented: What alternative explanations beyond consumer preference might account for the continued production of various Kit Kat flavors in Japan since 1973?

### Incorporate Real-World Scenarios
**Rule**: Tie the question to practical or observable scenarios to encourage applied reasoning.
**Example**:
Original: What can we infer about safety from the plane crash description?
Augmented: In a real-world aviation investigation scenario, how would the premise information about survival rates support or refute claims about the safety of children on the flight?

### Use Chain-of-Reasoning Prompts
**Rule**: Reformulate the question to require multi-step reasoning to reach the answer.
**Example**:
Original: Does the premise about technology news entail the hypothesis about Sunday worship?
Augmented: What cultural and historical knowledge must be applied to connect the mere mention of Sunday in a technology news context to the religious significance of the day, and how does this create an entailment relationship?

### Introduce Temporal Dynamics
**Rule**: Reformulate questions to explore how phenomena change over time.
**Example**:
Original: What does the Kit Kat example tell us about Japanese consumer preferences?
Augmented: How has the relationship between Kit Kat production and Japanese consumer preferences evolved since the product's 1973 introduction, and what does this evolution imply about the entailment in the current example?

### Integrate Explanatory Questions
**Rule**: Directly ask for an explanation of the reasoning behind an answer.
**Example**:
Original: Is it reasonable to conclude no children died in the crash?

Augmented: Explain the logical steps that connect the premise statement 'all passengers and crew have survived' to the entailment of 'no children were killed in the accident.'

## EI Question Augmentation Rules – ARC

### Convert Direct Questions into Why-Based Questions
**Rule**: Reformulate the question to ask about the reasons or mechanisms behind the original query.
**Example**:
Original: Does the premise entail that the day mentioned is important for religious worship?
Augmented: Why would Sunday being mentioned in the premise entail that it's a day for Christian worship?

### Emphasize Cause-Effect Relationships
**Rule**: Reframe the question to highlight causal relationships rather than surface-level descriptions.
**Example**:
Original: What does the premise state about the plane crash?
Augmented: How does the statement that all passengers and crew survived establish the entailment relationship with the hypothesis about no children being killed?

### Focus on Processes or Mechanisms
**Rule**: Augment questions to ask about the step-by-step processes involved in a phenomenon.
**Example**:
Original: How does the premise information about Kit Kat in Japan relate to the hypothesis?
Augmented: What is the inferential process by which we can conclude Japanese people like Kit Kat based on the premise about product variety and continued production since 1973?

### Include Counterfactual Scenarios
**Rule**: Reformulate the question to explore what would happen if a certain condition were different.
**Example**:
Original: Is the hypothesis about survival entailed by the premise?
Augmented: How would the entailment relationship change if the premise stated that 'most' rather than 'all' passengers survived the crash?

### Highlight Comparisons and Contrasts
**Rule**: Augment the question to compare different cases or conditions.
**Example**:
Original: What relationship exists between the premise about technology articles and the hypothesis about Sunday?
Augmented: How does the inference that Sunday is a Christian worship day differ from other possible inferences about Sunday mentioned in the technology news premise?

### Encourage Hypothesis Exploration
**Rule**: Reformulate the question to ask about possible explanations or hypotheses.
**Example**:
Original: Why is there an entailment between the Kit Kat production and Japanese preferences?
Augmented: What alternative explanations beyond consumer preference might account for the continued production of various Kit Kat flavors in Japan since 1973?

### Incorporate Real-World Scenarios
**Rule**: Tie the question to practical or observable scenarios to encourage applied reasoning.
**Example**:
Original: What can we infer about safety from the plane crash description?
Augmented: In a real-world aviation investigation scenario, how would the premise information about survival rates support or refute claims about the safety of children on the

flight?

### Use Chain-of-Reasoning Prompts
**Rule**: Reformulate the question to require multi-step reasoning to reach the answer.
**Example**:
Original: Does the premise about technology news entail the hypothesis about Sunday worship?
Augmented: What cultural and historical knowledge must be applied to connect the mere mention of Sunday in a technology news context to the religious significance of the day, and how does this create an entailment relationship?

### Introduce Temporal Dynamics
**Rule**: Reformulate questions to explore how phenomena change over time.
**Example**:
Original: What does the Kit Kat example tell us about Japanese consumer preferences?
Augmented: How has the relationship between Kit Kat production and Japanese consumer preferences evolved since the product's 1973 introduction, and what does this evolution imply about the entailment in the current example?

### Integrate Explanatory Questions
**Rule**: Directly ask for an explanation of the reasoning behind an answer.
**Example**:
Original: Is it reasonable to conclude no children died in the crash?
Augmented: Explain the logical steps that connect the premise statement 'all passengers and crew have survived' to the entailment of 'no children were killed in the accident.'

---

## EI Question Augmentation Rules – ANLI

### Convert Direct Questions into Why-Based Questions
**Rule**: Reformulate the question to ask about the reasons or mechanisms behind the original query.
**Example**:
Original: Does the premise entail that the day mentioned is important for religious worship?
Augmented: Why would Sunday being mentioned in the premise entail that it's a day for Christian worship?

### Emphasize Cause-Effect Relationships
**Rule**: Reframe the question to highlight causal relationships rather than surface-level descriptions.
**Example**:
Original: What does the premise state about the plane crash?
Augmented: How does the statement that all passengers and crew survived establish the entailment relationship with the hypothesis about no children being killed?

### Focus on Processes or Mechanisms
**Rule**: Augment questions to ask about the step-by-step processes involved in a phenomenon.
**Example**:
Original: How does the premise information about Kit Kat in Japan relate to the hypothesis?
Augmented: What is the inferential process by which we can conclude Japanese people like Kit Kat based on the premise about product variety and continued production since 1973?

### Include Counterfactual Scenarios
**Rule**: Reformulate the question to explore what would happen if a certain condition were different.
**Example**:
Original: Is the hypothesis about survival entailed by the premise?

Augmented: How would the entailment relationship change if the premise stated that 'most' rather than 'all' passengers survived the crash?

### Highlight Comparisons and Contrasts
**Rule**: Augment the question to compare different cases or conditions.
**Example**:
Original: What relationship exists between the premise about technology articles and the hypothesis about Sunday?
Augmented: How does the inference that Sunday is a Christian worship day differ from other possible inferences about Sunday mentioned in the technology news premise?

### Encourage Hypothesis Exploration
**Rule**: Reformulate the question to ask about possible explanations or hypotheses.
**Example**:
Original: Why is there an entailment between the Kit Kat production and Japanese preferences?
Augmented: What alternative explanations beyond consumer preference might account for the continued production of various Kit Kat flavors in Japan since 1973?

### Incorporate Real-World Scenarios
**Rule**: Tie the question to practical or observable scenarios to encourage applied reasoning.
**Example**:
Original: What can we infer about safety from the plane crash description?
Augmented: In a real-world aviation investigation scenario, how would the premise information about survival rates support or refute claims about the safety of children on the flight?

### Use Chain-of-Reasoning Prompts
**Rule**: Reformulate the question to require multi-step reasoning to reach the answer.
**Example**:
Original: Does the premise about technology news entail the hypothesis about Sunday worship?
Augmented: What cultural and historical knowledge must be applied to connect the mere mention of Sunday in a technology news context to the religious significance of the day, and how does this create an entailment relationship?

### Introduce Temporal Dynamics
**Rule**: Reformulate questions to explore how phenomena change over time.
**Example**:
Original: What does the Kit Kat example tell us about Japanese consumer preferences?
Augmented: How has the relationship between Kit Kat production and Japanese consumer preferences evolved since the product's 1973 introduction, and what does this evolution imply about the entailment in the current example?

### Integrate Explanatory Questions
**Rule**: Directly ask for an explanation of the reasoning behind an answer.
**Example**:
Original: Is it reasonable to conclude no children died in the crash?
Augmented: Explain the logical steps that connect the premise statement 'all passengers and crew have survived' to the entailment of 'no children were killed in the accident.'

---

**EI Question Augmentation Rules – CommonsenseQA**

**Convert Direct Questions into Why-Based Questions**
**Rule**: Reformulate the question to ask about the reasons or mechanisms behind the original query.
**Example**:
Original: Does the premise entail that the day mentioned is important for religious worship?
Augmented: Why would Sunday being mentioned in the premise entail that it's a day for Christian worship?

**Emphasize Cause-Effect Relationships**
**Rule**: Reframe the question to highlight causal relationships rather than surface-level descriptions.
**Example**:
Original: What does the premise state about the plane crash?
Augmented: How does the statement that all passengers and crew survived establish the entailment relationship with the hypothesis about no children being killed?

**Focus on Processes or Mechanisms**
**Rule**: Augment questions to ask about the step-by-step processes involved in a phenomenon.
**Example**:
Original: How does the premise information about Kit Kat in Japan relate to the hypothesis?
Augmented: What is the inferential process by which we can conclude Japanese people like Kit Kat based on the premise about product variety and continued production since 1973?

**Include Counterfactual Scenarios**
**Rule**: Reformulate the question to explore what would happen if a certain condition were different.
**Example**:
Original: Is the hypothesis about survival entailed by the premise?
Augmented: How would the entailment relationship change if the premise stated that 'most' rather than 'all' passengers survived the crash?

**Highlight Comparisons and Contrasts**
**Rule**: Augment the question to compare different cases or conditions.
**Example**:
Original: What relationship exists between the premise about technology articles and the hypothesis about Sunday?
Augmented: How does the inference that Sunday is a Christian worship day differ from other possible inferences about Sunday mentioned in the technology news premise?

**Encourage Hypothesis Exploration**
**Rule**: Reformulate the question to ask about possible explanations or hypotheses.
**Example**:
Original: Why is there an entailment between the Kit Kat production and Japanese preferences?
Augmented: What alternative explanations beyond consumer preference might account for the continued production of various Kit Kat flavors in Japan since 1973?

**Incorporate Real-World Scenarios**
**Rule**: Tie the question to practical or observable scenarios to encourage applied reasoning.
**Example**:
Original: What can we infer about safety from the plane crash description?
Augmented: In a real-world aviation investigation scenario, how would the premise information about survival rates support or refute claims about the safety of children on the

---

flight?

### Use Chain-of-Reasoning Prompts
**Rule**: Reformulate the question to require multi-step reasoning to reach the answer.
**Example**:
Original: Does the premise about technology news entail the hypothesis about Sunday worship?
Augmented: What cultural and historical knowledge must be applied to connect the mere mention of Sunday in a technology news context to the religious significance of the day, and how does this create an entailment relationship?

### Introduce Temporal Dynamics
**Rule**: Reformulate questions to explore how phenomena change over time.
**Example**:
Original: What does the Kit Kat example tell us about Japanese consumer preferences?
Augmented: How has the relationship between Kit Kat production and Japanese consumer preferences evolved since the product's 1973 introduction, and what does this evolution imply about the entailment in the current example?

### Integrate Explanatory Questions
**Rule**: Directly ask for an explanation of the reasoning behind an answer.
**Example**:
Original: Is it reasonable to conclude no children died in the crash?
Augmented: Explain the logical steps that connect the premise statement 'all passengers and crew have survived' to the entailment of 'no children were killed in the accident.'

---

## EI Question Augmentation Rules – Date

### Convert Direct Questions into Why-Based Questions
**Rule**: Reformulate the question to ask about the reasons or mechanisms behind the original query.
**Example**:
Original: Does the premise entail that the day mentioned is important for religious worship?
Augmented: Why would Sunday being mentioned in the premise entail that it's a day for Christian worship?

### Emphasize Cause-Effect Relationships
**Rule**: Reframe the question to highlight causal relationships rather than surface-level descriptions.
**Example**:
Original: What does the premise state about the plane crash?
Augmented: How does the statement that all passengers and crew survived establish the entailment relationship with the hypothesis about no children being killed?

### Focus on Processes or Mechanisms
**Rule**: Augment questions to ask about the step-by-step processes involved in a phenomenon.
**Example**:
Original: How does the premise information about Kit Kat in Japan relate to the hypothesis?
Augmented: What is the inferential process by which we can conclude Japanese people like Kit Kat based on the premise about product variety and continued production since 1973?

### Include Counterfactual Scenarios
**Rule**: Reformulate the question to explore what would happen if a certain condition were different.
**Example**:
Original: Is the hypothesis about survival entailed by the premise?

Augmented: How would the entailment relationship change if the premise stated that 'most' rather than 'all' passengers survived the crash?

### Highlight Comparisons and Contrasts
**Rule**: Augment the question to compare different cases or conditions.
**Example**:
Original: What relationship exists between the premise about technology articles and the hypothesis about Sunday?
Augmented: How does the inference that Sunday is a Christian worship day differ from other possible inferences about Sunday mentioned in the technology news premise?

### Encourage Hypothesis Exploration
**Rule**: Reformulate the question to ask about possible explanations or hypotheses.
**Example**:
Original: Why is there an entailment between the Kit Kat production and Japanese preferences?
Augmented: What alternative explanations beyond consumer preference might account for the continued production of various Kit Kat flavors in Japan since 1973?

### Incorporate Real-World Scenarios
**Rule**: Tie the question to practical or observable scenarios to encourage applied reasoning.
**Example**:
Original: What can we infer about safety from the plane crash description?
Augmented: In a real-world aviation investigation scenario, how would the premise information about survival rates support or refute claims about the safety of children on the flight?

### Use Chain-of-Reasoning Prompts
**Rule**: Reformulate the question to require multi-step reasoning to reach the answer.
**Example**:
Original: Does the premise about technology news entail the hypothesis about Sunday worship?
Augmented: What cultural and historical knowledge must be applied to connect the mere mention of Sunday in a technology news context to the religious significance of the day, and how does this create an entailment relationship?

### Introduce Temporal Dynamics
**Rule**: Reformulate questions to explore how phenomena change over time.
**Example**:
Original: What does the Kit Kat example tell us about Japanese consumer preferences?
Augmented: How has the relationship between Kit Kat production and Japanese consumer preferences evolved since the product's 1973 introduction, and what does this evolution imply about the entailment in the current example?

### Integrate Explanatory Questions
**Rule**: Directly ask for an explanation of the reasoning behind an answer.
**Example**:
Original: Is it reasonable to conclude no children died in the crash?
Augmented: Explain the logical steps that connect the premise statement 'all passengers and crew have survived' to the entailment of 'no children were killed in the accident.'

---

### EI Question Augmentation Rules – Math

### Convert Direct Questions into Why-Based Questions
**Rule**: Reformulate the question to ask about the reasons or mechanisms behind the original query.
**Example**:
Original: Does the premise entail that the day mentioned is important for religious worship?
Augmented: Why would Sunday being mentioned in the premise entail that it's a day for

Christian worship?

**Emphasize Cause-Effect Relationships**

**Rule**: Reframe the question to highlight causal relationships rather than surface-level descriptions.

**Example**:

Original: What does the premise state about the plane crash?

Augmented: How does the statement that all passengers and crew survived establish the entailment relationship with the hypothesis about no children being killed?

**Focus on Processes or Mechanisms**

**Rule**: Augment questions to ask about the step-by-step processes involved in a phenomenon.

**Example**:

Original: How does the premise information about Kit Kat in Japan relate to the hypothesis?

Augmented: What is the inferential process by which we can conclude Japanese people like Kit Kat based on the premise about product variety and continued production since 1973?

**Include Counterfactual Scenarios**

**Rule**: Reformulate the question to explore what would happen if a certain condition were different.

**Example**:

Original: Is the hypothesis about survival entailed by the premise?

Augmented: How would the entailment relationship change if the premise stated that 'most' rather than 'all' passengers survived the crash?

**Highlight Comparisons and Contrasts**

**Rule**: Augment the question to compare different cases or conditions.

**Example**:

Original: What relationship exists between the premise about technology articles and the hypothesis about Sunday?

Augmented: How does the inference that Sunday is a Christian worship day differ from other possible inferences about Sunday mentioned in the technology news premise?

**Encourage Hypothesis Exploration**

**Rule**: Reformulate the question to ask about possible explanations or hypotheses.

**Example**:

Original: Why is there an entailment between the Kit Kat production and Japanese preferences?

Augmented: What alternative explanations beyond consumer preference might account for the continued production of various Kit Kat flavors in Japan since 1973?

**Incorporate Real-World Scenarios**

**Rule**: Tie the question to practical or observable scenarios to encourage applied reasoning.

**Example**:

Original: What can we infer about safety from the plane crash description?

Augmented: In a real-world aviation investigation scenario, how would the premise information about survival rates support or refute claims about the safety of children on the flight?

**Use Chain-of-Reasoning Prompts**

**Rule**: Reformulate the question to require multi-step reasoning to reach the answer.

**Example**:

Original: Does the premise about technology news entail the hypothesis about Sunday worship?

Augmented: What cultural and historical knowledge must be applied to connect the mere mention of Sunday in a technology news context to the religious significance of the day, and how does this create an entailment relationship?

**Introduce Temporal Dynamics**

**Rule**: Reformulate questions to explore how phenomena change over time.

**Example**:

Original: What does the Kit Kat example tell us about Japanese consumer preferences?

Augmented: How has the relationship between Kit Kat production and Japanese consumer preferences evolved since the product's 1973 introduction, and what does this evolution imply about the entailment in the current example?

**Integrate Explanatory Questions**
**Rule**: Directly ask for an explanation of the reasoning behind an answer.
**Example**:
Original: Is it reasonable to conclude no children died in the crash?
Augmented: Explain the logical steps that connect the premise statement 'all passengers and crew have survived' to the entailment of 'no children were killed in the accident.'

---

### EI Question Augmentation Rules – StrategyQA

**Convert Direct Questions into Why-Based Questions**
**Rule**: Reformulate the question to ask about the reasons or mechanisms behind the original query.
**Example**:
Original: Does the premise entail that the day mentioned is important for religious worship?
Augmented: Why would Sunday being mentioned in the premise entail that it's a day for Christian worship?

**Emphasize Cause-Effect Relationships**
**Rule**: Reframe the question to highlight causal relationships rather than surface-level descriptions.
**Example**:
Original: What does the premise state about the plane crash?
Augmented: How does the statement that all passengers and crew survived establish the entailment relationship with the hypothesis about no children being killed?

**Focus on Processes or Mechanisms**
**Rule**: Augment questions to ask about the step-by-step processes involved in a phenomenon.
**Example**:
Original: How does the premise information about Kit Kat in Japan relate to the hypothesis?
Augmented: What is the inferential process by which we can conclude Japanese people like Kit Kat based on the premise about product variety and continued production since 1973?

**Include Counterfactual Scenarios**
**Rule**: Reformulate the question to explore what would happen if a certain condition were different.
**Example**:
Original: Is the hypothesis about survival entailed by the premise?
Augmented: How would the entailment relationship change if the premise stated that 'most' rather than 'all' passengers survived the crash?

**Highlight Comparisons and Contrasts**
**Rule**: Augment the question to compare different cases or conditions.
**Example**:
Original: What relationship exists between the premise about technology articles and the hypothesis about Sunday?
Augmented: How does the inference that Sunday is a Christian worship day differ from other possible inferences about Sunday mentioned in the technology news premise?

**Encourage Hypothesis Exploration**
**Rule**: Reformulate the question to ask about possible explanations or hypotheses.
**Example**:
Original: Why is there an entailment between the Kit Kat production and Japanese preferences?
Augmented: What alternative explanations beyond consumer preference might account for the continued production of various Kit Kat flavors in Japan since 1973?

**Incorporate Real-World Scenarios**
**Rule**: Tie the question to practical or observable scenarios to encourage applied reasoning.
**Example**:
Original: What can we infer about safety from the plane crash description?
Augmented: In a real-world aviation investigation scenario, how would the premise information about survival rates support or refute claims about the safety of children on the

flight?

**Use Chain-of-Reasoning Prompts**
**Rule**: Reformulate the question to require multi-step reasoning to reach the answer.
**Example**:
Original: Does the premise about technology news entail the hypothesis about Sunday worship?
Augmented: What cultural and historical knowledge must be applied to connect the mere mention of Sunday in a technology news context to the religious significance of the day, and how does this create an entailment relationship?

**Introduce Temporal Dynamics**
**Rule**: Reformulate questions to explore how phenomena change over time.
**Example**:
Original: What does the Kit Kat example tell us about Japanese consumer preferences?
Augmented: How has the relationship between Kit Kat production and Japanese consumer preferences evolved since the product's 1973 introduction, and what does this evolution imply about the entailment in the current example?

**Integrate Explanatory Questions**
**Rule**: Directly ask for an explanation of the reasoning behind an answer.
**Example**:
Original: Is it reasonable to conclude no children died in the crash?
Augmented: Explain the logical steps that connect the premise statement 'all passengers and crew have survived' to the entailment of 'no children were killed in the accident.'

---

## EI Question Augmentation Rules – TabMWP

**Convert Direct Questions into Why-Based Questions**
**Rule**: Reformulate the question to ask about the reasons or mechanisms behind the original query.
**Example**:
Original: Does the premise entail that the day mentioned is important for religious worship?
Augmented: Why would Sunday being mentioned in the premise entail that it's a day for Christian worship?

**Emphasize Cause-Effect Relationships**
**Rule**: Reframe the question to highlight causal relationships rather than surface-level descriptions.
**Example**:
Original: What does the premise state about the plane crash?
Augmented: How does the statement that all passengers and crew survived establish the entailment relationship with the hypothesis about no children being killed?

**Focus on Processes or Mechanisms**
**Rule**: Augment questions to ask about the step-by-step processes involved in a phenomenon.
**Example**:
Original: How does the premise information about Kit Kat in Japan relate to the hypothesis?
Augmented: What is the inferential process by which we can conclude Japanese people like Kit Kat based on the premise about product variety and continued production since 1973?

**Include Counterfactual Scenarios**
**Rule**: Reformulate the question to explore what would happen if a certain condition were different.
**Example**:
Original: Is the hypothesis about survival entailed by the premise?
Augmented: How would the entailment relationship change if the premise stated that 'most' rather than 'all' passengers survived the crash?

**Highlight Comparisons and Contrasts**
**Rule**: Augment the question to compare different cases or conditions.

**Example**:
Original: What relationship exists between the premise about technology articles and the hypothesis about Sunday?
Augmented: How does the inference that Sunday is a Christian worship day differ from other possible inferences about Sunday mentioned in the technology news premise?

**Encourage Hypothesis Exploration**
**Rule**: Reformulate the question to ask about possible explanations or hypotheses.
**Example**:
Original: Why is there an entailment between the Kit Kat production and Japanese preferences?
Augmented: What alternative explanations beyond consumer preference might account for the continued production of various Kit Kat flavors in Japan since 1973?

**Incorporate Real-World Scenarios**
**Rule**: Tie the question to practical or observable scenarios to encourage applied reasoning.
**Example**:
Original: What can we infer about safety from the plane crash description?
Augmented: In a real-world aviation investigation scenario, how would the premise information about survival rates support or refute claims about the safety of children on the flight?

**Use Chain-of-Reasoning Prompts**
**Rule**: Reformulate the question to require multi-step reasoning to reach the answer.
**Example**:
Original: Does the premise about technology news entail the hypothesis about Sunday worship?
Augmented: What cultural and historical knowledge must be applied to connect the mere mention of Sunday in a technology news context to the religious significance of the day, and how does this create an entailment relationship?

**Introduce Temporal Dynamics**
**Rule**: Reformulate questions to explore how phenomena change over time.
**Example**:
Original: What does the Kit Kat example tell us about Japanese consumer preferences?
Augmented: How has the relationship between Kit Kat production and Japanese consumer preferences evolved since the product's 1973 introduction, and what does this evolution imply about the entailment in the current example?

**Integrate Explanatory Questions**
**Rule**: Directly ask for an explanation of the reasoning behind an answer.
**Example**:
Original: Is it reasonable to conclude no children died in the crash?
Augmented: Explain the logical steps that connect the premise statement 'all passengers and crew have survived' to the entailment of 'no children were killed in the accident.'

## O  THE USE OF LARGE LANGUAGE MODELS (LLMS)

To enhance clarity and readability, we utilized OpenAI GPT-5 and Google Gemini 2.5-Pro exclusively as a language polishing tool. Its role was confined to proofreading, grammatical correction, and stylistic refinement—functions analogous to those provided by traditional grammar checkers and dictionaries. This tool did not contribute to the generation of new scientific content or ideas, and its usage is consistent with standard practices for manuscript preparation

