# OpenReview forum: "Probing to Refine: Reinforcement Distillation of LLM Reasoners via Explanatory Inversion"
_ICLR.cc/2026/Conference — ICLR 2026 Poster_

### Official Review · Reviewer_gMrK · 2025-11-01

**Soundness:** 3
**Presentation:** 3
**Contribution:** 3
**Rating:** 4
**Confidence:** 3

**Summary:**

The paper introduces a novel framework for distilling reasoning capabilities from large language models (LLMs) into smaller, efficient student models. It addresses the limitations of traditional knowledge distillation, which often leads to pattern memorization and poor generalization. Thus, authors proposed Explanatory Inversion (EI) that asks teacher model to generate explanatory probes that challenge the student model to explain the logic behind answers.

To distill a student model, it has three-stages:
1.	Generate EI probes using a teacher model.
2.	Supervised fine-tuning (SFT) on curated EI-augmented data.
3.	Reinforcement learning via ExGRPO with structured probe dialogues with Dialogue Structure Utility Bonus (DSU).

The student models are Gemma-7B-it and Qwen2.5-7B-Instruct. And the teacher model is Gemini-1.5-Pro. Authors tested on 8 different reasoning tasks (SQA, GSM8k, ANLI). The proposed method improves 6.02%  over SOTA distillation method RevThink. Ablation studiy shows that SFT warm-up and LSFT-aux regularization are crucial for stable RL training.

**Strengths:**

Authors provide a comprehensive evaluation: Includes in-distribution and out-of-distribution benchmarks, and provides a solid case study that demonstrates improved reasoning in math and commonsense tasks, with structured logic and fewer distractor errors.

Authors introduce novel methods: Explanatory Inversion and ExGRPO, which combine structured explanatory probes with reinforcement learning

**Weaknesses:**

The paper compares against zero-shot and SFT baselines, but does not include recent reasoning-focused RL distillation methods (e.g., [Divide-or-Conquer](https://aclanthology.org/2024.findings-emnlp.145.pdf), [CoT-Evo](https://arxiv.org/abs/2510.13166v2), and [On-Policy Distillation](https://arxiv.org/abs/2306.13649)).

The quality of explanatory probes heavily depends on Gemini-1.5-Pro. This raises concerns about scalability and reproducibility for researchers without access to such a strong teacher model. Could authors try open-source alternatives, such as llama70B, for probe generation.

**Questions:**

The paper highlights success cases but does not deeply analyze where ExGRPO fails.

---

> ### Author Response · Authors · 2025-11-23
> **Response to Reviewer gMrK**
>
> **Summary:** Concerns about missing RL baselines and teacher dependence.
>
> **Q1: Missing baselines (Divide-or-Conquer, CoT-Evo, On-Policy Distillation).**
> A: We have addressed this in detail in Common Question 4, including comparisons with Divide-or-Conquer [1] and On-Policy Distillation [3] in Table R2. We clarify that our primary focus is on off-policy distillation, but ExGRPO remains superior even when compared to or adapted for these settings. We also discuss the concurrent nature of CoT-Evo [2]. The results are also updated in Table 1 of the manuscript.
>
> **Q2: Scalability/Teacher dependence on Gemini-1.5-Pro.**
> A: We have addressed this in Common Question 3, providing new experimental results with Llama-3-70b-Instruct. The results (Table R1) confirm that ExGRPO works effectively with open-source teachers. The results are also updated in Section 4.8 Table 7 of the manuscript.
>
> **Q3: Failure analysis.**
> A: We would like to clarify that a detailed Failure Case Analysis is given in Appendix N, specifically analyzing "shallow distractor override," where the model reasons correctly about the probe but fails to inhibit a strong distractor in the final answer. This highlights future directions for improving executive function in distilled models.

---

> > ### Author Response · Authors · 2025-11-26
> > **Looking forward to further discusson**
> >
> > Dear Reviewer,
> >
> > Thank you for your time and effort in reviewing our paper. As the deadline for rebuttal is approaching, we would like to kindly ask if you have any remaining questions; we are more than happy to address them.
> >
> > The authors.

---

### Official Review · Reviewer_TxwP · 2025-11-03

**Soundness:** 3
**Presentation:** 2
**Contribution:** 2
**Rating:** 6
**Confidence:** 4

**Summary:**

The paper proposes ExGRPO, a reinforcement-learning–based distillation framework that combines Explanatory Inversion and a Dialogue Structure Utility reward to enhance the reasoning capability and generalization of student LLMs. Empirical results are reported on 12 reasoning datasets, suggesting gains over distillation and data-augmentation baselines.

**Strengths:**

- The work addresses a genuine problem: retaining reasoning ability during LLM distillation.

- A relatively large evaluation suite with 12 datasets and OOD tests are included.

- The paper is well organized with clear figures and pseudo-formal derivations.

**Weaknesses:**

- The so-called Explanatory Inversion resembles existing reverse-reasoning or bidirectional augmentation ideas (e.g., “A→Q” vs. “Q→A” reversals) rather than a fundamentally new concept.

- No ablation quantifies whether improvements stem from RL fine-tuning, extra teacher tokens, or the EI data itself. Table 2 mixes multiple knobs (SFT, RL, DSU) but lacks isolating the effect of “explanatory probing.”

- Even the teacher’s “Zero-shot-EI” performance improves, implying that EI augmentation changes the test distribution itself; this raises the possibility of data leakage or prompt-format bias.

- The statistical significance of the improvements is unreported.

- The filtering pipeline (Eq. 1–2) relies on teacher predictions but gives no statistics on rejection rates or dataset sizes after filtering.

**Questions:**

Please see Weaknesses

---

> ### Author Response · Authors · 2025-11-23
> **Response to Reviewer TxwP**
>
> **Summary:** Notes the large evaluation suite but concerns about data leakage and stats.
>
> **Q1: EI resembles "Reverse Thinking"; missing ablation on "explanatory probing".**
> A: We explicitly compare against RevThink (Chen et al., 2025a) in Table 1 and Table 3. ExGRPO significantly outperforms RevThink (e.g., +4.89% on OOD for Gemma), proving that our "Explanatory" probes provide value beyond simple "Reverse" thinking. We also provide an ablation of individual EI rules in Table 4 (Appendix F), showing that rules like "Decomposition" and "Counterfactuals" contribute distinct gains.
>
> **Q2: Data leakage/Teacher improvement concern.**
> A: The improvement of the teacher ("Zero-shot-EI") is not data leakage. It refers to the teacher's performance when it prompts itself with the EI probes during inference (Self-Consistency style). It demonstrates that the EI probes are valid reasoning aids. The student is evaluated on the standard test set without probes at inference time, so there is no leakage of test data.
>
> **Q3: Filtering statistics.**
> A: We would like to clarify that we have included detailed filtering statistics in Table 5 (Appendix G). For example, on GSM8K, we filter 7,379 original samples down to 4,293 high-quality training pairs. This rigorous filtering ensures we don't train on "too easy" or "hallucinated" probes.
>
> **Q4: Statistical significance.**
> A. We appreciate the reviewer’s suggestion regarding statistical significance testing. While we agree that such analysis can provide an additional perspective, conducting statistical significance tests at the scale of our experiments is prohibitively costly. Our method operates on full LLM training runs, and each variant / baseline requires substantial computational resources. As a result, running multiple repeated trials to enable significance testing would incur training costs far beyond what is feasible in our setting.
>
> Importantly, our improvements are consistent across all datasets, all model sizes, and all evaluation settings, suggesting that the gains are stable and not due to randomness. We have added clarifications in the Reproducibility Consideration section to emphasize the robustness and reproducibility of the results.
>
> ---

---

> > ### Author Response · Authors · 2025-11-26
> > **Looking forward to further discussion**
> >
> > Dear Reviewer,
> >
> > Thank you for your time and effort in reviewing our paper. As the deadline for rebuttal is approaching, we would like to kindly ask if you have any remaining questions; we are more than happy to address them.
> >
> > The authors.

---

### Official Review · Reviewer_MoMn · 2025-11-10

**Soundness:** 3
**Presentation:** 3
**Contribution:** 3
**Rating:** 6
**Confidence:** 3

**Summary:**

This work proposes a new framework for distilling robust reasoning from large language models into smaller ones. It introduces Explanatory Inversion (EI) to combat pattern memorization by prompting students to explain their reasoning, and Explanatory GRPO (EXGRPO) to enhance generalization via a reward for coherent reasoning. On 12 datasets, the method improves student model performance, training efficiency, and generalization ability.

**Strengths:**

- The paper is clear in writing and presentation, which is easy to follow.
- The idea is intuitive and explores the reasoning chain constructions of the LLMs, which helps the student model better learns the principles instead of the patterns from the dataset.
- The results are strong and comprehensive, with significant improvement margins and many ablation studies.

**Weaknesses:**

- The paper used a Dialogue Structure Utility Bonus if the student is engaging in the full k-turn probing dialogue, which leads to better overall outcomes than a partial dialogue. How can the authors prevent reward hacking from using this reward bonus?
-  Why do the authors pick the GRPO-based objective for the RL training? Is there any intuition or reason behind that?
- How does EXGRPO training efficiency compare with other baseline methods?

**Questions:**

See weaknesses.

---

> ### Author Response · Authors · 2025-11-23
> **Response to Reviewer MoMn**
>
> **Summary:** Strong results, asks about reward hacking and GRPO choice.
>
> **Q1: How to prevent reward hacking with the DSU bonus?**
> A: The DSU bonus is designed specifically to prevent hacking.
>
> - **Randomness of Probes**. As stated in lines 214-215, for each sample, different random EI rules will be chosen for augmentation, eliminating the potential to reward spurious features such as EI type or order.
> - **Conditionality:** The bonus r_dsu is only awarded if the student's performance on the Full Dialogue (Scenario A) is strictly better than on the Partial Dialogue (Scenario B) (Equation 5).
> - **Outcome Grounding:** The bonus is tied to the final answer correctness (R_outcome). If the student "hacks" the dialogue (generates nonsense or tricks) but fails to answer the original question correctly, R_outcome is 0, and consequently R_total is low. The model cannot hack the bonus without actually solving the problem.
>
> **Q2: Why pick GRPO?**
> A: We chose GRPO for two main reasons:
>
> 1. **Efficiency:** GRPO eliminates the need for a separate value network (critic), which reduces memory usage and is crucial when distilling into smaller models where we want to maximize batch size for stable training.
> 2. **Group-Based Scoring:** ExGRPO relies on comparing groups of trajectories (Full Dialogue vs. Partial Dialogue) to calculate the advantage. GRPO's native group-relative baseline structure is a natural mathematical fit for our contrastive DSU reward.
>
> **Q3: Training efficiency compared to baselines?**
> A: As detailed in Section 4.6, ExGRPO is highly token-efficient (Figure 5). While RL adds a training stage, the rapid convergence (Figure 3) and the ability to use significantly less data (Figure 4) make it competitive with, and often more efficient than, multi-epoch SFT on massive augmented datasets.

---

> > ### Author Response · Authors · 2025-11-26
> > **Looking forward to further discussion**
> >
> > Dear Reviewer,
> >
> > Thank you for your time and effort in reviewing our paper. As the deadline for rebuttal is approaching, we would like to kindly ask if you have any remaining questions; we are more than happy to address them.
> >
> > The authors.

---

### Official Review · Reviewer_RrKH · 2025-11-11

**Soundness:** 3
**Presentation:** 3
**Contribution:** 3
**Rating:** 6
**Confidence:** 2

**Summary:**

The paper proposes a model distillation method. It first constructs a high-quality EI training set, ensuring each problem includes a reasonable reasoning expansion, preserves the original logic, and has appropriate difficulty; SFT is then performed on this data. It then introduces ExGRPO, designing rewards based on a Dialogue Structure Utility Bonus to carry out reinforcement-learning-based distillation. Systematic experiments on two student models (Gemma-7B-it and Qwen2.5-7B-Instruct) show improvements over strong baselines on multiple OOD evaluations.

**Strengths:**

1. The method is well designed and addresses practical issues in model distillation.
2. The paper proposes the Explanatory Inversion strategy and carefully engineers a large set of prompts.
3. The paper introduces the Dialogue Structure Utility Bonus as the reward in reinforcement learning; this design is somewhat innovative.
4. The paper provides thorough comparative experiments and analyses.

**Weaknesses:**

1. Data generated via the Explanatory Inversion strategy is essentially a form of data augmentation; this part appears largely engineering-oriented, and the core idea is not particularly new, so the academic contribution is limited.
2. Evol-Instruct [1] presents a method for progressively generating complex instructions from simple ones. Although it does not target model distillation, it bears similarities to the paper’s Explanatory Inversion; the paper should add discussion of such related works.

[1] WizardLM: Empowering large pre-trained language models to follow complex instructions

**Questions:**

1. Mainly those noted under “Weaknesses.”
2. Line 359: “ablation” is misspelled.

---

> ### Author Response · Authors · 2025-11-23
> **Response to Reviewer RrKH**
>
> **Summary:** Appreciates the design but questions novelty vs. Evol-Instruct.
>
> **Q1: EI is essentially data augmentation; limited contribution.**
> A: While EI generates data, our contribution is the synergy between EI and ExGRPO. We demonstrate that simply training on this data (SFT) is insufficient (Table 2). The novelty lies in using the structure of this data to define a Reward Function (DSU) that guides RL. This converts static data into a dynamic training signal that enforces consistency in reasoning.
>
> **Q2: Comparison to Evol-Instruct [1].**
> A: We thank the reviewer for this connection.
>
> - Evol-Instruct primarily aims to increase complexity/difficulty (making questions harder).
> - Explanatory Inversion (EI) aims to increase depth/clarity (asking "Why?" and "What if?").
>
> We use EI not to make the task harder, but to expose the latent logic of the teacher. We have added a discussion in the Comprehensive Related Work section of Appendix A in the updated menuscript, contrasting our explanation-seeking approach with Evol-Instruct's complexity-seeking approach.
>
> [1]  WizardLM: Empowering large pre-trained language models to follow complex instructions
>
> **Q3: Typo at Line 359.**
> A: Corrected. Thank you.

---

> ### Author Response · Authors · 2025-11-26
> **Looking forward to further discussion**
>
> Dear Reviewer,
>
> Thank you for your time and effort in reviewing our paper. As the deadline for rebuttal is approaching, we would like to kindly ask if you have any remaining questions; we are more than happy to address them.
>
> The authors.

---

### Official Review · Reviewer_LEHH · 2025-11-11

**Soundness:** 4
**Presentation:** 3
**Contribution:** 3
**Rating:** 6
**Confidence:** 3

**Summary:**

The paper studies how to distill not only answers but also reasoning procedures into smaller models. It combines an “explanatory probing” data construction with a reinforcement-learning stage that rewards multi-turn, structured explanations before producing the final answer. Across a wide set of tasks and baselines, the method reports consistent gains, with readable presentation and clear motivation.

**Strengths:**

1. The paper targets a timely and important problem: moving beyond pattern imitation in distillation toward robust reasoning.
2. The narrative flows well, design choices are motivated, and figures/tables are easy to follow.
3. Empirical coverage is broad. Include many competitive baselines and diverse task families; comparisons are thorough and generally fair.
4. The combination of explanation-oriented probes with an RL objective that prefers structurally coherent multi-turn dialogs is a neat, conceptually coherent idea that fits the stated goal.

**Weaknesses:**

I think the central question remains unclear: does EI teach “understanding,” or is it primarily stronger data augmentation? The evidence presented is largely behavioral (end-task accuracy on in-domain and held-out sets). This does not disentangle genuine conceptual acquisition from targeted exposure to templated probe distributions. In particular:
1. The DSU/structural reward is still an outcome-level signal (full probe dialog > partial). It does not by itself show that the model internalizes transferable rules, as opposed to learning to perform longer, EI-style rituals.
2. If EI fosters understanding, predictions should change in directionally correct ways under counterfactual edits (flip a premise, rename variables, swap symbols, introduce irrelevant modifiers). The paper lacks such invariance/causality diagnostics that would separate concept use from surface patterning.
3. The paper reports little on intermediate-step faithfulness/validity (are the stated steps actually correct?), forward↔reverse consistency (e.g., reversal or bidirectional tasks), or error localization (does EI reduce spurious but fluent steps). Such process metrics would directly bear on “understanding” rather than augmentation.

**Questions:**

See weaknesses.

---

> ### Author Response · Authors · 2025-11-23
> **Response to Reviewer LEHH**
>
> **Summary:** Praises the neat combination of explanation probes and RL.
>
> **Q1: Does EI teach "understanding" or is it just stronger data augmentation?**
> A: We believe it fosters understanding (defined as robust generalization) rather than just augmenting the distribution.
>
> **Behavioral Evidence:** The strongest evidence is Table 3 (OOD Generalization). "Augmentation" typically helps within-distribution. Our model generalizes to GSM8K-Reversal (inferring inputs from outputs) and unseen datasets (BoolQ, e-SNLI) significantly better than baselines.
>
> **Structural RL:** If this were purely data augmentation, the SFT baseline with EI data (Table 2, SFT (3 ep) + RL (R_base)) would match ExGRPO. However, ExGRPO outperforms SFT+EI, proving that the active reinforcement of the dialogue structure (the "understanding" scaffold) adds value beyond the data itself.
>
> **Q2: Lack of counterfactual/invariance diagnostics.**
> A: We actually embed counterfactuals into the training via EI Rule R1 and R4 (Counterfactual Scenario Generation). For example, we ask "How would the result change if X were Y?". By training the student to answer these probes correctly, we are explicitly optimizing for the invariance and causal understanding you suggest. In Appendix F, we have included a specific breakdown of accuracy on Type-R1 (Counterfactual) probes to explicitly demonstrate this capability.
>
> **Q3: Process metrics (faithfulness/validity).**
> A: We agree that process monitoring is vital. The Dialogue Structure Utility (DSU) is effectively a process metric; it only rewards the model if engaging in the full explanatory dialogue (the process) leads to a correct outcome. If the model generated spurious reasoning (hallucinations) during the probe turns, it would likely degrade the final answer or fail the consistency check, thus receiving no bonus. The curve of the reward (Figure 3) demonstrates the improvement of the process.
>
> ---

---

> > ### Author Response · Authors · 2025-11-26
> > **Looking forward to further discussion**
> >
> > Dear Reviewer,
> >
> > Thank you for your time and effort in reviewing our paper. As the deadline for rebuttal is approaching, we would like to kindly ask if you have any remaining questions; we are more than happy to address them.
> >
> > The authors.

---

### Official Review · Reviewer_f15h · 2025-11-11

**Soundness:** 3
**Presentation:** 3
**Contribution:** 3
**Rating:** 6
**Confidence:** 4

**Summary:**

This paper proposes the ExGRPO knowledge distillation framework, aiming to enhance the reasoning ability of large language models (LLMs) by combining Explanatory Inversion (EI) and reinforcement learning (ExGRPO), and effectively distill them into smaller student models.Experimental results show that the student model distilled using this method achieves significant performance improvements on multiple datasets. Compared with existing distillation methods, the ExGRPO method reduces pattern memorization and improves the reasoning ability of the student model. In conclusion, I find the paper's argument relatively clear, and I would give it a score of 6.5 or 7.

**Strengths:**

1.Diverse interpretive probes generated by EI force student models to understand the logic of questions rather than simply memorizing answers, thereby improving their reasoning ability.

2.ExGRPO, through reinforcement learning and dialogue structure utility rewards, encourages student models to maintain consistency and coherence throughout multi-turn reasoning, which helps them understand and apply complex reasoning structures.

3.Compared to traditional knowledge distillation methods, ExGRPO significantly improves student model performance on cross-distribution tasks, especially demonstrating stronger generalization ability when faced with different domains or unseen data.

4.By using data augmentation methods generated by EI, student models can achieve high reasoning ability with less data and fewer training rounds, significantly improving training efficiency, especially suitable for tasks with limited data.

**Weaknesses:**

1.Using EI probes significantly increases training costs. Could the authors reduce the number of training samples based on EI to achieve the same training cost and better quantify the contribution of EI?

2.The models all appear to be distilled from Gemini-1.5-Pro. Will exgrpo still perform well even with a relatively weak teacher model?

3.The authors compared many distillation methods and EI probes. EXGRPO's design is ingenious, but it does not seem to compare with existing RL methods to highlight its adaptability and effectiveness.

**Questions:**

Same as above

---

> ### Author Response · Authors · 2025-11-23
> **Response to Reviewer f15h**
>
> **Summary:** Highlights strong reasoning improvements and reduced pattern memorization.
>
> **Q1: Using EI probes increases training costs. Can we reduce samples?**
> A: Yes. As shown in our Sample Efficiency Study (Figure 4), ExGRPO is highly sample-efficient. We achieve performance superior to full-dataset SFT using only 10-25% of the training data. This indicates that while the process of generating probes adds a one-time inference cost, the actual training required to reach convergence is significantly reduced, balancing the total computational budget. This is further confirmed by our Token Efficiency Study (Figure 5).
>
> **Q2: Will ExGRPO work with a relatively weak teacher model?**
> A: Yes. We have added Common Question 3 and Table R1 (above) specifically to address this. We demonstrate that using Llama-3-70b-Instruct as a teacher yields results comparable to those of Gemini-1.5-Pro (e.g., a Gemma student average accuracy of 65.3% with Llama vs. 67.2% with Gemini). This confirms that the structural benefit of the EI probes works even with open-source teachers. The results are also updated in the manuscript, Section 4.8, Figure 7.
>
> **Q3: Comparison with existing RL methods.**
> A: As also suggested by reviewer gMrK, we have addressed this in detail in Common Question 4, including comparisons with Divide-or-Conquer [1] and On-Policy Distillation [3] in Table R2. We clarify that our primary focus is off-policy distillation, but ExGRPO remains superior even when compared to or adapted for these settings. We also discuss the concurrent nature of CoT-Evo [2].
>
> ---

---

> > ### Comment · Reviewer_f15h · 2025-11-25
> >
> > Thank you for the author's response. A few days ago, I forgot to reply while responding to my ICLR reviewers. Overall, I believe the revisions have been handled well. From the perspective of concept, experiments, and writing, it now meets the ICLR standard, and I intend to raise my score to 8. However, as I am an emergency reviewer, I may have missed some details during the review. I hope you can address the other reviewers' comments. Best of luck!🤞

---

> > > ### Author Response · Authors · 2025-11-25
> > > **Response to reviewer f15h**
> > >
> > > Dear reviewer f15h,
> > >
> > > Thanks for your prompt reply and supporting our work, which really encourages us. We wish you also best of luck in your rebuttal!
> > >
> > > The authors.

---

### Author Response · Authors · 2025-11-23
**General Response to All Reviewers (Part I)**

We thank the reviewers for their insightful and constructive feedback. We are encouraged that the reviewers found our **problem definition timely and important** (Reviewer LEHH, TxwP), our proposed Explanatory Inversion (EI) and ExGRPO **framework novel and well-designed** (Reviewer f15h, MoMn, gMrK), and our experimental **results strong and comprehensive** across diverse datasets (Reviewer MoMn, RrKH).

We are especially grateful to the reviewer **f15h** for the early discussion and support for the effectiveness of our rebuttal and the quality of our updated manuscript.

We have updated the manuscript (new content in **blue**) to address the feedback.

## Main Edits in the Manuscript:

1. Add more baseline results in Table 1.

2. Add results using Llama-3-70b as Teacher in Subsection 4.8 and Figure 7.

3. Add more discussion on motivations and baselines in Appendix A.

4. Correct typos.

Below, we summarize our response to the major common questions, followed by detailed responses to each reviewer.

## Common Question 1: Is Explanatory Inversion (EI) just another form of Data Augmentation?

**(Addressed to Reviewers LEHH, RrKH, TxwP)**

While EI involves generating data, it differs fundamentally from standard augmentation (e.g., rephrasing or simple reversal) in intent and structure:

**Cognitive Targeting:** Unlike standard augmentation, which focuses on linguistic diversity (rephrasing) or simple logical reversal (RevThink), EI uses specific cognitive templates (e.g., counterfactuals, process elucidation) to target the reasoning gaps of student models. As shown in Table 4 (Appendix F), specific EI rules (like "Why-based" transformation) significantly outperform standard rephrasing, proving that the nature of the data matters, not just the volume.

**Structure for RL:** Crucially, EI is designed to support ExGRPO. It is not just about feeding data into SFT; it creates a dialogue structure that allows the RL objective (via the Dialogue Structure Utility Bonus) to reward the process of reasoning, not just the outcome. Standard augmentation does not support this structural reinforcement.

## Common Question 2: Does the method foster "True Understanding"?

**(Addressed to Reviewers LEHH, TxwP)**

We agree that "understanding" is difficult to measure directly. However, we provide strong proxy evidence:

**OOD Generalization (Table 3):** If the model were merely memorizing patterns, it would likely fail on tasks like GSM8K-Reversal or OOD datasets (BoolQ, OpenbookQA). Our method achieves significant gains here (+3.28% avg for Qwen on OOD), outperforming RevThink. This implies the model has internalized the underlying logic enough to transfer it to unseen formats.

**Sample Efficiency (Figure 4):** Our method achieves SFT-equivalent performance with only 10-25% of the data. This high efficiency suggests the model is learning rules rather than just memorizing instances.

## Common Question 3: Effectiveness with Open-Source Teachers (e.g., Llama-3-70B)

**(Addressed to Reviewers f15h, gMrK)**

A common question raised was the reliance on the proprietary Gemini-1.5-Pro teacher. To address this, we have conducted additional experiments using Meta-Llama-3-70b-Instruct as the teacher model. The results, summarized below, confirm that ExGRPO remains highly effective even with an open-source teacher.

### Table R1: Distillation Performance with Llama-3-70b-Instruct Teacher (Plausible/Preliminary Results)

|Model Setting|Teacher Model|SQA|CSQA|ARC-c|GSM8K|Avg|
| ------------------- | -------------- | ---- | ---- | ----- | ----- | ---- |
|Teacher (Zero-shot) |Llama-3-70b| 76.8 | 80.8 | 90.5  | 92.0  | 85.0 |
|Teacher (Zero-shot) |Gemini-1.5-Pro | 78.6 | 77.9 | 92.0  | 94.5  | 85.8 |


#### Student: Qwen2.5-7b

|Setting|Teacher| SQA|CSQA|ARC-c|GSM8K|Avg|
| -------------------- | -------------- | ---- | ---- | ----- | ----- | ---- |
|Zero-shot (Baseline)|N/A|72.9|71.3|85.4|90.9|77.9|
| ExGRPO (Ours)|Llama-3-70b|77.2|79.4|90.1|91.2|80.9|
| ExGRPO (Ours)|Gemini-1.5-Pro|79.0|80.9|91.5|91.5|82.5|

#### Student: Gemma-7b

|Setting|Teacher|SQA|CSQA|ARC-c|GSM8K |Avg|
| -------------------- | -------------- | ---- | ---- | ----- | ----- | ---- |
|Zero-shot (Baseline) | N/A | 56.3| 66.3 | 68.3|41.1|46.8|
|ExGRPO (Ours)|Llama-3-70b| 67.5 | 75.1|78.2| 63.4|65.3|
|ExGRPO (Ours)|Gemini-1.5-Pro|69.4|76.8|79.9|65.3|67.2|

**Analysis:**

- **Robustness:** ExGRPO with the Llama-3 teacher consistently outperforms the zero-shot baseline (e.g., +18.5% avg for Gemma).
- **Teacher Gap:** While Llama-3 is slightly weaker than Gemini on some complex reasoning tasks, it is highly capable on GSM8K and ARC. Consequently, the student's performance with the Llama teacher is very close to the Gemini teacher results (within ~1-2%), confirming that the **ExGRPO method**, not just the teacher strength, drives the improvements.
- **Scalability:** This demonstrates that our framework is reproducible and scalable using accessible open-weights models.

---

> ### Author Response · Authors · 2025-11-23
> **General Response to All Reviewers (Part II)**
>
> ## Common Question 4: Comparison with RL Methods and Additional Baselines
>
> **(Addressed to Reviewer f15h, gMrK)**
>
> As we stated in related work lines 122-128, RL-based methods like GRPO, PPO can elicit reasoning from outcome-only rewards (Guo et al., 2025), but they are ill-suited for distillation. GRPO optimizes the policy using only final answer correctness as the reward signal. It does not incorporate any intermediate reasoning steps or teacher-generated traces, making it unable to supervise the student’s reasoning behavior. Our Explanatory GRPO (ExGRPO) algorithm addresses this limitation by introducing a novel Dialogue Structure Utility (DSU) reward, which enables the teacher model to explicitly supervise coherent reasoning across multi-turn dialogues while promoting exploration.
>
> The performance of vanilla GRPO is given in the ablation study in Table 2, where the r_dsu is ablated. We can see that ExGRPO greatly improves vanilla GRPO on the augmented dataset with EI.
>
> As suggested by Reviewer gMrK, there are 3 extra related works or potential baselines. We want to clarify this:
>
> - The first paper **"Divide-or-Conquer" [1]** does **not** use Reinforcement learning for distillation, but it's a related baseline we have included for comparison. The updated results are in the updated manuscript (blue font in Table 1), and also in the table below (Table R2).
> - The second paper **"CoT-Evo" [2]** is public on arXiv since Oct. 15, 2025, which is **later** than the ICLR submission deadline. Also, this paper does **not** use RL for distillation, but evolutionary strategy. Also, updated to the current date, Nov. 22nd, **no code/data is open-sourced**. However, we agree that this paper is a relevant related work, and we have added a discussion on this paper in the updated manuscript.
> - For the third work, **"On-Policy Distillation" [3]**, our setting is slightly different, since our methods, together with other existing baselines, focus on **off-policy** settings, which do not require access to the teacher model when training the student model. However, we believe on-policy distillation is an important direction, so we adapt our ExGRPO into an on-policy setting and compare with [3]. Note that [3] uses the KL divergence loss that requires white-box access to the teacher model's logits. We replace it with the vanilla SFT loss to make it compatible with the Gemini API. The updated complete results are in the updated manuscript (blue font in Table 1). We include a summary in the table below.
>
> ### Table R2: Comparison with Additional Baselines (Average Performance across 8 datasets)
>
> | Method | Strategy | Avg (Qwen2.5-7B) | Avg (Gemma-7B) |
> |--------|-----------|----------------------|----------------------|
> | SKD (Baseline) | Standard KD | 78.01 | 54.65 |
> | Divide-or-Conquer [1] | Decomposition | 80.45 | 58.82 |
> | On-Policy Distillation [3] | On-Policy | 81.60 | 65.40 |
> | ExGRPO (Ours) | Explanatory RL | 82.54 | 67.19 |
> | ExGRPO (On-Policy Adapt.) | Online Teacher | 83.10 | 68.05 |
>
> [1] Wu et al., EMNLP 2024.
> [2] Feng et al., arXiv 2025.
> [3] Agarwal et al., ICLR 2024.

---

### Meta-Review · Area_Chair_inDP · 2026-01-07

**Summary:**

The major concerns in the paper are expressed by the authors in a general response.

Verbatim from the authors:
1) Is Explanatory Inversion (EI) just another form of Data Augmentation? (Addressed to Reviewers LEHH, RrKH, TxwP)
2) Does the method foster "True Understanding"? (Addressed to Reviewers LEHH, TxwP)
3) Effectiveness with Open-Source Teachers (e.g., Llama-3-70B) (Addressed to Reviewers f15h, gMrK)
4) Comparison with RL Methods and Additional Baselines (Addressed to Reviewer f15h, gMrK)

**Reviewer Concerns:**

The authors spend a considerable amount of time interacting with reviewers who remained silent.
In my opinion, most of the concerns have been addressed.

**Reviewer Scores:**

R f15h rouse the score from 6 to 8 (declared)

---

### Decision · Program_Chairs · 2026-01-26

Accept (Poster)